# Provably Efficient Black-Box Action Poisoning Attacks Against Reinforcement Learning

**Guanlin Liu, Lifeng Lai**
Department of Electrical and Computer Engineering
University of California, Davis
{glnliu,lflai}@ucdavis.edu

## Abstract

Due to the broad range of applications of reinforcement learning (RL), understanding the effects of adversarial attacks against RL model is essential for the safe applications of this model. Prior works on adversarial attacks against RL mainly focus on either observation poisoning attacks or environment poisoning attacks. In this paper, we introduce a new class of attacks named action poisoning attacks, where an adversary can change the action signal selected by the agent. Compared with existing attack models, the attacker's ability in the proposed action poisoning attack model is more restricted, and hence the attack model is more practical. We study the action poisoning attack in both white-box and black-box settings. We introduce an adaptive attack scheme called LCB-H, which works for most RL agents in the black-box setting. We prove that the LCB-H attack can force any efficient RL agent, whose dynamic regret scales sublinearly with the total number of steps taken, to choose actions according to a policy selected by the attacker very frequently, with only sublinear cost. In addition, we apply LCB-H attack against a popular model-free RL algorithm: UCB-H. We show that, even in the black-box setting, by spending only logarithm cost, the proposed LCB-H attack scheme can force the UCB-H agent to choose actions according to the policy selected by the attacker very frequently.

## 1 Introduction

Reinforcement learning (RL), a framework of control-theoretic problem that makes decisions over time under uncertain environment, has many applications in a variety of scenarios such as recommendation systems [Zhao et al., 2018], autonomous driving [O' Kelly et al., 2018], finance [Liu et al., 2020] and business management [Nazari et al., 2018], to name a few. As RL models are being increasingly used in safety critical and security related applications, it is critical to developing trustworthy RL systems. Understanding the effects of adversarial attacks on RL systems is the first step towards the goal of safe applications of RL models.

While there is much existing work addressing adversarial attacks on supervised learning models [Szegedy et al., 2014, Goodfellow et al., 2015, Kurakin et al., 2017, Moosavi-Dezfooli et al., 2017, Wang et al., 2018, Cohen et al., 2019, Dohmatob, 2019, Wang et al., 2019, Zhang and Zhu, 2019, Carmon et al., 2019, Pinot et al., 2019, Alayrac et al., 2019, Dasgupta et al., 2019, Cicalese et al., 2020], the understanding of adversarial attacks on RL models is less complete. Among the limited existing works on adversarial attacks against RL, they formally or experimentally considers different types of poisoning attack [Behzadan and Munir, 2017, Huang and Zhu, 2019, Ma et al., 2019, Zhang et al., 2020, Sun et al., 2021, Rakhsha et al., 2020, 2021]. [Sun et al., 2021] discusses the differences between the poisoning attacks. In the observation poisoning attack setting, the attacker is able to manipulate the observations of the agent. Before the agent receives the reward signal or the state

35th Conference on Neural Information Processing Systems (NeurIPS 2021).

signal from the environment, the attacker is able to modify the data. In the environment poisoning setting, the attacker could directly change the underlying environment, i.e., the Markov decision process (MDP) model.

In this paper, we introduce a suite of novel attacks on RL named action poisoning attacks. In the proposed action poisoning attacks models, an attacker sits between the agent and the environment and could change the agent's action. For example, in auto-driving systems, the attacker could implement destabilizing forces or manipulate the action signal, so as to change the brake force. Compared with the observation poisoning or environment poisoning attacks, the ability of the attacker in the action poisoning attack is more restricted, which brings some design challenges. In particular, compared with observation poisoning and environment poisoning attacks, the effects of the action poisoning attack on the change of observation is less direct. Furthermore, when the action space is discrete and finite, the ability of the action poisoning attacker is severely limited. We note that the goal of this paper is not to promote action manipulation attacks. Our goal is to understand the potential risks of action manipulation attacks. Understanding the risks of different kinds of adversarial attacks on RL is essential for the safe applications of RL model and designing robust RL systems.

In this paper, we investigate action poisoning attacks in both white-box and black-box settings. The white-box attack setting makes strong assumptions. In particular, the attacker has full information of the underlying MDP, the agent's algorithm or the agent's previous policy models, or all of them. While it is often unrealistic to exactly know the underlying environment or have the right to obtain the information of the agent's model, the understanding of the white-box attacks could provide insights on how to design black-box attack schemes. In the black-box setting, the attacker has no prior information of the underlying MDP and does not know the agent's algorithm. The only information the attacker has is observations generated from the environment when the agent interacts with the environment. The black-box setting is much more practical and is suitable for more realistic scenarios.

**Contributions:** Our main contributions are as follows: (1) We propose an action poisoning attack model in which the attacker aims to force the agent to learn a policy selected by the attacker (will be called target policy in the sequel) by changing the agent's actions to other actions. We use loss and cost functions to evaluate the effects of the action poisoning attack on a RL agent. The cost is the cumulative number of times when the attacker changes the agent's action, and the loss is the cumulative number of times when the agent does not follow the target policy. It is clearly of interest to minimize both the cost and loss functions. (2) In the white-box setting, we introduce an attack strategy named $\alpha$-portion attack. We show that the $\alpha$-portion attack strategy can force any sub-linear-regret RL agent to choose actions according to the target policy specified by the attacker with sub-linear cost and sub-linear loss. (3) We develop a black-box attack strategy, LCB-H, that nearly matches the performance of the white-box $\alpha$-portion attack. To the best of our knowledge, LCB-H is the first black-box action poisoning attack scheme that provably works against RL agents. (4) We investigate the impact of the LCH-B attack on UCB-H [Jin et al., 2018], a popular and efficient model-free $Q$-learning algorithm, and show that, by spending only logarithm cost, the LCB-H attack can force the UCB-H agent to choose actions according to the target policy with logarithm loss.

**Related work:** Existing works on poisoning attacks against RL have studied different types of adversarial manipulations. [Ma et al., 2019] studies reward poisoning attack against batch RL in which the attacker is able to gather and modify the collected batch data. [Rakhsha et al., 2020] proposes a white-box environment poisoning model in which the attacker could manipulate the original MDP to a poisoned MDP. [Behzadan and Munir, 2017, Zhang et al., 2020] study online white-box reward poisoning attacks in which the attacker could manipulate the reward signal before the agent receives it. [Sun et al., 2021] proposes a practical black-box poisoning algorithm called VA2C-P. Their empirical results show that VA2C-P works for deep policy gradient RL agents without any prior knowledge of the environment. [Rakhsha et al., 2021] develops a black-box reward poisoning attack strategy called U2, that can provably attack any efficient RL algorithms. There are also some interesting works that focus on attacking multi-arm bandit problems. In particular, [Jun et al., 2018, Liu and Shroff, 2019] investigate reward poisoning attacks and show that the attacker can manipulate the behavior of the bandit algorithms by spending only logarithm cost. [Guan et al., 2020] considers a reward poisoning attack model where an adversary attacks with a certain probability at each round.[Liu and Lai, 2020] proposes an action poisoning attack strategy against multi-arm bandit problems. Existing work on action poisoning attacks against RL is limited. There are some empirical studies in deep RL [Pinto et al., 2017, Lee et al., 2020, Sun et al., 2021].

**Potential negative societal impacts:** The attack strategies discussed in this paper could potentially be used by malicious users to attack real RL systems. The goal of this paper is to understand and highlight the potential negative consequences of action manipulation attacks, with the hope to raise awareness of this issue in the research community. It is also our intention to design robust RL algorithms that can defend against such attacks in our future work.

## 2  Problem Formulation

Consider a tabular episodic MDP $\mathcal{M} = (\mathcal{S}, \mathcal{A}, H, P, R)$, where $\mathcal{S}$ is the state space with $|\mathcal{S}| = S$, $\mathcal{A}$ is the action space with $|\mathcal{A}| = A$, $H \in \mathbb{Z}^+$ is the number of steps in each episode, $P_h : \mathcal{S} \times \mathcal{A} \times \mathcal{S} \to [0,1]$ is the probability transition function which maps state-action-state pair to a probability, $R_h : \mathcal{S} \times \mathcal{A} \to [0,1]$ represents the reward function in the step $h$. In this paper, the probability transition functions and the reward functions can be different at different steps.

The agent interacts with the environment in a sequence of episodes. The total number of episodes is $K$. In each episode $k \in [K]$ of this MDP, the initial states $s_1$ is generated randomly by a distribution or chosen by the environment. Initial states may be different between episodes. At each step $h \in [H]$ of an episode, the agent observes the state $s_h$ and chooses an action $a_h$. After receiving the action, the environment generates a random reward $r_h \in [0,1]$ derived from a distribution with mean $R_h(s_h, a_h)$ and next state $s_{h+1}$ which is drawn from the distribution $P_h(\cdot|s_h, a_h)$. $P_h(\cdot|s,a)$ represents the probability distribution over states if action $a$ is taken for state $s$. The agent stops interacting with environment after $H$ steps and starts another episode.

The policy $\pi$ of the agent is expressed as a mappings $\pi : \mathcal{S} \times [H] \times \mathcal{A} \to [0,1]$. $\pi_h(a|s)$ represents the probability of taking action $a$ in state $s$ under stochastic policy $\pi$ at step $h$. We have that $\sum_{a \in \mathcal{A}} \pi_h(a|s) = 1$. A deterministic policy is a policy that maps each state to a particular action. For notation convenience, for a deterministic policy $\pi$, we use $\pi_h(s)$ to denote the action $a$ which satisfies $\pi_h(a|s) = 1$. Interacting with the environment $\mathcal{M}$, the policy induces a random trajectory $\{s_1, a_1, r_1, s_2, a_2, r_2, \cdots, s_H, a_H, r_H, s_{H+1}\}$.

We use $V_h^\pi : \mathcal{S} \to \mathbb{R}$ to denote the value function at step $h$ under policy $\pi$. Given a policy $\pi$ and step $h$, the value function of a state $s \in \mathcal{S}$ and the $Q$-function $Q_h^\pi : \mathcal{S} \times \mathcal{A} \to \mathbb{R}$ of a state-action pair $(s,a)$ are defined as: $V_h^\pi(s) = \mathbb{E}\left[\sum_{h'=h}^H r_{h'}|s_h = s, \pi\right]$ and $Q_h^\pi(s,a) = \mathbb{E}\left[\sum_{h'=h}^H r_{h'}|s_h = s, a_h = a, \pi\right]$, which represent the expected total rewards received from step $h$ to $H$, under policy $\pi$, starting from state $s$ and state-action pair $(s,a)$ respectively. It is well-known that the value function and $Q$-function satisfy the Bellman consistency equations. For notation simplicity, we denote $V_{H+1}^\pi = \mathbf{0}$, $Q_{H+1}^\pi = \mathbf{0}$ and $P_h V_{h+1}^\pi(s,a) = \mathbb{E}_{s' \sim P_h(\cdot|s,a)}[V_{h+1}^\pi(s')]$.

In this paper, we assume that the state space $\mathcal{S}$ and action space $\mathcal{A}$ are finite sets, and the planning horizon $H$ is finite. Reward $r_h$ is bounded by $[0,1]$, so the value function and $Q$-function are bounded. Under this case, there always exists an optimal policy $\pi^*$ such that $\pi^*$ maximize the value function and $Q$-function: $V_h^*(s) := V_h^{\pi^*}(s) = \sup_\pi V_h^\pi(s)$ and $Q_h^*(s,a) := Q_h^{\pi^*}(s,a) = \sup_\pi Q_h^\pi(s,a)$, for all $s$, $a$ and $h$. We measure the performance of the agent over $K$ episodes by the regret defined as:

$$\text{Regret}(K) = \sum_{k=1}^K [V_1^*(s_1^k) - V_1^{\pi^k}(s_1^k)], \tag{1}$$

where $s_1^k$ is the initial state and $\pi^k$ is the control policy followed by the agent for each episode $k$.

In this paper, we introduce a novel adversary setting, in which the attacker sits between the agent and the environment. The attacker can monitor the state, the actions of the agent and the reward signals from the environment. Furthermore, the attacker can introduce action poisoning attacks on RL agent. In particular, at each episode $k$ and step $h$, after the agent chooses an action $a_h^k$, the attacker can change it to another action $\widetilde{a}_h^k \in \mathcal{A}$. If the attacker decides not to attack, $\widetilde{a}_h^k = a_h^k$. Then the environment receives $\widetilde{a}_h^k$, and generates a random reward $r_h^k$ with mean $R_h(s_h^k, \widetilde{a}_h^k)$ and the next state $s_{h+1}^k$ which is drawn from the distribution $P_h(\cdot|s_h^k, \widetilde{a}_h^k)$. The agent and attacker receive the reward $r_h^k$ and the next state $s_{h+1}^k$ from the environment. Note that the agent does not know the attacker's manipulations and the presence of the attacker and hence will still view $r_h^k$ as the reward and $s_{h+1}^k$ as the next state generated from state-action pair $(s_h^k, a_h^k)$.

The attacker has a target policy $\pi^\dagger$. We assume that the target policy $\pi^\dagger$ is a deterministic policy. The attacker's goal is to manipulate the agent into following the target policy $\pi^\dagger$ to pick its actions. We measure the performance of the attack over $K$ episodes by the total attack cost and the total number of the steps that the agent does not follow the target policy $\pi^\dagger$. By setting $\mathbb{1}(\cdot)$ as the indicator function, the attack cost function and the loss function are defined as

$$\text{Cost}(K, H) = \sum_{k=1}^{K} \sum_{h=1}^{H} \mathbb{1}\left(\widetilde{a}_h^k \neq a_h^k\right), \quad \text{Loss}(K, H) = \sum_{k=1}^{K} \sum_{h=1}^{H} \mathbb{1}\left(a_h^k \neq \pi^\dagger(s_h^k)\right), \quad (2)$$

The attacker aims to minimize both the attack cost and the loss of attacks, or minimize one of them subject to a constraint on the one another. However, obtaining optimal solutions to these optimization problems is challenging. As the first step towards understanding the impact of action poisoning attacks, we design some specific simple yet effective attack strategies.

## 3 Attack Strategy and Analysis

In this paper, we study the black-box action poisoning attack problem. In black-box attack case, the attacker has no prior knowledge about the underlying environment and the agent's policy. It only knows the observations, i.e., $s_h^k$, $a_h^k$, and $r_h^k$, generated when the agent interacts with the environment. This makes the attack practical as the attacker only needs to hijack the communication between the environment and the agent without stealing information from or attacking the agent and the environment. To build up intuitions about the proposed black-box action poisoning attack strategy, we first consider a white-box attack model, in which the attacker knows the underlying MDP and hence it is easier to design attack schemes. Building on insights obtained from the white-box attack schemes, we then introduce our proposed black-box attack strategy and analyze its performance.

### 3.1 White-box Attack

In the white-box attack model, the attacker has full information of the underlying MDP $\mathcal{M}$. Thus, the attacker is able to calculate $V_h^*(s)$ and $Q_h^*(s, a)$ according to the Bellman optimality equations:

$$Q_h^*(s, a) = R_h(s, a) + P_h V_{h+1}^*(s, a), \quad V_h^*(s) = \max_{a \in \mathcal{A}} Q_h^*(s, a). \quad (3)$$

Since $V_{H+1}^\pi = \mathbf{0}$ and $Q_{H+1}^\pi = \mathbf{0}$, $V_h^*(s)$ and $Q_h^*(s, a)$ can be obtained from the Bellman optimality equation. The optimal policy $\pi^*$ are derived from $\pi_h^*(s) = \arg\max_{a \in \mathcal{A}} Q_h^*(s, a)$.

With the knowledge of the optimal policy, the attacker can perform an intuitive attack: exchange the optimal action and the target action. In particular, at the step $h$ and state $s$, when the agent picks the optimal action $a = \pi_h^*(s)$, the attacker changes it to the action specified by the target policy $\widetilde{a} = \pi_h^\dagger(s)$. When the agent selects the target action $a = \pi_h^\dagger(s)$, the attacker changes it to the action might be taken under the optimal policy $\widetilde{a} = \pi_h^*(s)$. In addition, when the agent's action does not follow the optimal policy or the target policy, the attacker does not attack. We name this attack scheme as the exchange attack (E-attack) strategy. From the agent's point of view, $\pi^\dagger$ becomes the optimal strategy under the E-attack strategy, as the agent does not know the presence of the attacker. If the optimal policy is singular, any RL algorithm with sub-linear regret will learn to follow the optimal strategy in his observation, i.e. $\pi^\dagger$, with a sub-linear regret. As the result, the loss will be sub-linear. However, the cost of the E-attack strategy may be up to $\mathcal{O}(T)$, where $T = KH$ is the total number of steps. The main reason is that, in the E-attack strategy, the attacker needs to change the actions whenever the agent chooses an action specified by the target policy $\pi^\dagger$, which happens most of the time as the agent views $\pi^\dagger$ as the optimal policy. Furthermore, another drawback of the E-attack is that the expected reward the agent receives is not impacted.

Even though the E-attack strategy discussed above could force the agent to follow the target policy $\pi^\dagger$, the cost is too high and it does not affect the agent's total expected rewards. In order to reduce the cost and have real impact on the agent's total expected rewards, the attacker should avoid to attack when the agent takes an action specified by $\pi^\dagger$. This is possible if $\pi^\dagger$ satisfies certain conditions to be specified in the sequel.

Before presenting the proposed attack strategy, we first discuss conditions under which such an attack is possible. If the target policy is the worst policy such that $V_h^{\pi^\dagger}(s) = \inf_\pi V_h^\pi(s)$, the attacker

can not force the agent to learn the target policy without attacking the target action. For notation simplicity, we denote $V_h^\dagger(s) := V_h^{\pi^\dagger}(s)$. The combination of the attacker and the environment can be considered as a new environment to the agent. As the action poisoning attack only changes the actions, it can impact but does not have direct control of the agent's observations. Although the action poisoning attack is widely applicable, the attacker's ability is weaker than the attacker in the environment poisoning attack model. It is reasonable to limit the choice of the target policy. In this paper, we study a class of target policies denoted as $\Pi^\dagger$, for which each element $\pi^\dagger \in \Pi^\dagger$ satisfies

$$V_h^\dagger(s) > \min_{a \in \mathcal{A}} Q_h^\dagger(s, a), \tag{4}$$

for all state $s$ and all step $h$. That is $\pi^\dagger$ is not the worst policy.

**Assumption 1.** *For the underlying MDP $\mathcal{M} = (\mathcal{S}, \mathcal{A}, H, P, R)$, $\Pi^\dagger \neq \emptyset$, and the attacker's target policy $\pi^\dagger$ satisfies $\pi^\dagger \in \Pi^\dagger$.*

For a given target policy $\pi^\dagger$, as $|\mathcal{S}|$, $|\mathcal{A}|$ and $H$ are finite, the minimum of $Q_h^\dagger(s, a)$ subject to $a \in \mathcal{A}$ exists for all step $h$ and state $s$. We define the minimum gap $\Delta_{min}$ by

$$\Delta_{min} = \min_{h \in [H], s \in \mathcal{S}} \left( V_h^\dagger(s) - \min_{a \in \mathcal{A}} Q_h^\dagger(s, a) \right). \tag{5}$$

Under Assumption 1, the minimum gap is positive, i.e. $\Delta_{min} > 0$. This positive gap provides a chance of efficient action poisoning attacks. All results in this paper are based on Assumption 1.

We now introduce an effective white-box attack schemes: $\alpha$-portion attack. Specifically, at the step $h$ and state $s$, if the agent picks the target action, i.e., $a = \pi_h^\dagger(s)$, the attacker does not attack, i.e. $\widetilde{a} = a = \pi_h^\dagger(s)$. If the agent picks a non-target action, i.e., $a \neq \pi_h^\dagger(s)$, the $\alpha$-portion attack sets $\widetilde{a}$ as

$$\widetilde{a} = \begin{cases} \pi_h^\dagger(s), \text{with probability } 1 - \alpha \\ \arg\min_{a \in \mathcal{A}} Q_h^\dagger(s, a), \text{with probability } \alpha. \end{cases} \tag{6}$$

For a given target policy $\pi^\dagger$, we define $\pi_h^-(s) = \arg\min_{a \in \mathcal{A}} Q_h^\dagger(s, a)$. We have the following result:

**Lemma 1.** *If the attacker follows the $\alpha$-portion attack scheme on an RL agent, in the observation of the agent, the target policy $\pi^\dagger$ is the optimal policy.*

The detailed proof can be found in the Appendix B.1 of the supplementary material. Using Lemma 1, we can derive upper bounds of the loss and the cost functions of the $\alpha$-portion attack scheme.

**Theorem 1.** *Assume the expected regret $Regret(K)$ of the RL agent's algorithm is bounded by a sub-linear bound $\mathcal{R}(T)$, i.e., $Regret(K) \leq \mathcal{R}(T)$. The $\alpha$-portion attack will force the agent to learn the target policy $\pi^\dagger$ with the expected cost and the expected loss bounded by*

$$\mathbb{E}[\text{Cost}(K, H)] \leq \mathbb{E}[\text{Loss}(K, H)] \leq \mathcal{R}(T)/(\alpha \Delta_{min}), \tag{7}$$

*In addition, with probability $1 - p$, the loss and the cost is bounded by*

$$\text{Cost}(K, H) \leq \text{Loss}(K, H) \leq \left( \mathcal{R}(T) + 2H^2 \sqrt{\log(1/p)\mathcal{R}(T)} \right)/(\alpha \Delta_{min}). \tag{8}$$

The detailed proofs can be found in the Appendix B.2 of the supplementary material. In the white-box setting, the attacker can simply choose $\alpha = 1$ to most effectively attack RL agents. Intuitively speaking, if $\alpha = 1$, whenever the agent chooses a non-target action, the attacker changes it to the worst action under policy $\pi^\dagger$, so that all non-target actions become worse than the target action and the target policy becomes optimal in the observation of the agent.

### 3.2 Black-box Attack

In the black-box attack setting, the attacker has no prior information about the underlying environment and the agent's algorithm, it only observes the samples generated when the agent interacts with the environment. Since the $\alpha$-portion attack described in (6) for the white-box setting relies on the information of the underlying environment to solve $\pi_h^-$, the $\alpha$-portion attack is not applicable in the

black-box setting. However, by collecting the observations and evaluating the $Q$-function $Q_h^\dagger(s, a)$, the attacker can perform an attack to approximate the $\alpha$-portion attack. In the proposed attack scheme, the attacker evaluates the $Q$-values of the target policy $\pi^\dagger$ with an important sampling (IS) estimator. Then, the attacker calculates the lower confidence bound (LCB) on the $Q$-values so that he can explore and exploit the worst action by the LCB method. In this paper, we use Hoeffding-type martingale concentration inequalities to build the confidence bound. Using this information, the attacker can then carry out an attack similar to the $\alpha$-portion attack. We name the proposed attack strategy as LCB-H attack. The algorithm is summarized in Algorithm 1. In the following, we will show that the LCB-H attack nearly matches the performance of the $\alpha$-portion attack.

---

**Algorithm 1:** LCB-H attack strategy on RL algorithm

---

**Input:**
    Target policy $\pi^\dagger$.
1: Initialize $L_h(s, a) = -\infty$, $\hat{Q}_h^\dagger(s, a) = 0$, and $N_h(s, a) = 0$ for all state $s \in \mathcal{S}$, all action $a \in \mathcal{A}$ and all step $h \in [H]$.
2: **for** episode $k = 1, 2, \ldots, K$ **do**
3:     Receive $s_1^k$. Initialize the set of trajectory $traj = \{s_1^k\}$.
4:     **for** step $h = 1, 2, \ldots, H$ **do**
5:         The agent chooses an action $a_h^k$.
6:         **if** $a_h^k = \pi_h^\dagger(s_h^k)$ **then**
7:             The attacker does not attack, i.e. $\widetilde{a}_h^k = a_h^k$, and sets the IS weight $w_h = 1$.
8:         **else**
9:             
$$\widetilde{a}_h^k = \begin{cases} \underset{a \neq \pi_h^\dagger(s_h^k)}{\arg\min} L_h(s_h^k, a) \text{ and set } w_h = 0, & \text{with probability } 1/H, \\ \pi_h^\dagger(s_h^k) \text{ and set } w_h = H/(H-1), & \text{with probability } 1 - 1/H. \end{cases}$$
10:         **end if**
11:         The environment receives action $\widetilde{a}_h^k$, and returns the reward $r_h^k$ and the next state $s_{h+1}^k$.
12:         Update the trajectory by plugging $\widetilde{a}_h^k, r_h^k$ and $s_{h+1}^k$ into $traj$.
13:     **end for**
14:     Set the cumulative reward $G_{H+1} = 0$ and the importance ratio $\rho_{H+1:H+1} = 1$.
15:     **for** step $h = H, H-1, \ldots, 1$ **do**
16:         $G_h = r_h^k + G_{h+1}, \rho_{h:H+1} = \rho_{h+1:H+1} \cdot w_h, t = N_h(s_h^k, \widetilde{a}_h^k) \leftarrow N_h(s_h^k, \widetilde{a}_h^k) + 1$.
17:         $\hat{Q}_h^\dagger(s_h^k, \widetilde{a}_h^k) \leftarrow (1 - \frac{1}{t})\hat{Q}_h^\dagger(s_h^k, \widetilde{a}_h^k) + \frac{1}{t}\left(r_h^k + G_{h+1} \cdot \rho_{h+1:H+1}\right)$.
18:         $L_h(s_h^k, \widetilde{a}_h^k) = \hat{Q}_h^\dagger(s_h^k, \widetilde{a}_h^k) - (e(H-h) + 1)\sqrt{2\log(2SAT/p)/t}$.
19:     **end for**
20: **end for**

---

Here, we highlight the main idea of the LCB-H attack. As discussed in Section 3.1, if the attacker has full information of the MDP and knows the worst action of any state $s$ at any step $h$ under policy $\pi^\dagger$, he can simply change the agent's non-target action to the worst action. However, in the black-box setting, the attacker does not know the worst actions under policy $\pi^\dagger$. One intuitive idea is to estimate $Q^\dagger$ and find the possible worst actions by the estimates of $Q^\dagger$. Once the attacker obtains esimate $\hat{Q}^\dagger$, it carries out an attack similar to the $\alpha$-portion attack by setting $\alpha = 1/H$ (the reason why we set $\alpha = 1/H$ will be discussed in the sequel): 1) when the agent picks a target action, the LCB-H attacker does not attack ; 2) when the agent picks a non-target action, with probability $1 - \frac{1}{H}$, the LCB-H attacker changes it to the target action, while with probability $\frac{1}{H}$, the LCB-H attacker changes it to the action that has the lowest lower confidence bound value. Here, we use the lower confidence bound value because in the black-box setting, the LCB-H attacker does not know which action is the worst, and hence uses confidence bounds to explore and exploit the worst action.

As shown in Algorithm 1, after collecting observations, the LCB-H attacker uses IS estimator to evaluate the target policy, which is an off-policy method [Precup, 2000, Thomas et al., 2015]. The IS estimator provides an unbiased estimate of the target policy $\pi^\dagger$. Suppose $\pi^k$ is the control policy followed by the agent at episode $k$ and the attacker applies LCB-H attack on the agent. From Algorithm 1, the probability of an action $\widetilde{a}$ chosen by the behavior policy $b_h^k$ at the state $s$ and step $h$

can be written as

$$\mathbb{P}(\widetilde{a}|s, b_h^k) = \begin{cases} 1 & \text{if } \widetilde{a} = a_h^k = \pi_h^\dagger(s), \\ 1/H & \text{if } \widetilde{a} = \underset{a \neq \pi_h^\dagger(s)}{\arg\min} L_h^k(s,a) \text{ and } a_h^k \neq \pi_h^\dagger(s), \\ 1 - 1/H & \text{if } \widetilde{a} = \pi_h^\dagger(s) \text{ and } a_h^k \neq \pi_h^\dagger(s), \\ 0 & \text{if otherwise.} \end{cases} \quad (9)$$

Then, the trajectory at episode $k$, $\{s_1^k, \widetilde{a}_1^k, r_1^k, s_2^k, \widetilde{a}_2^k, r_2^k, \cdots, s_H^k, \widetilde{a}_H^k, r_H^k, s_{H+1}^k\}$, is generated under the behavior policy $b^k$. Since we assume that the target policy is a deterministic function, we have

$$\mathbb{P}(\widetilde{a}|s, \pi_h^\dagger) = \mathbb{1}(\widetilde{a} = \pi_h^\dagger(s)). \quad (10)$$

The importance sampling ratio $\rho_{h:H}^k = \prod_{h'=h}^H \frac{\mathbb{P}(\widetilde{a}_{h'}^k|s_{h'}^k, \pi^\dagger)}{\mathbb{P}(\widetilde{a}_{h'}^k|s_{h'}^k, b^k)}$ can be computed using (9) and (10), which is also used in Algorithm 1. Define the cumulative reward as $G_{h:H}^k = \sum_{h'=h}^H r_h^k$. For notation simplicity, we set $\rho_{h:H+1}^k = \rho_{h:H}^k$ and $G_{h:H+1}^k = G_{h:H}^k$ when $1 \leq h \leq H$, and $\rho_{H+1:H+1}^k = 1$ and $G_{H+1:H+1}^k = 0$. Since the trajectory is generated by following the behavior policy $b^k$, we have that for all step $h$ with $1 \leq h \leq H+1$, $V_h^{b^k}(s) = \mathbb{E}[G_{h:H}^k|s_h^k = s]$ and $V_h^\dagger(s) = \mathbb{E}[\rho_{h:H}^k G_{h:H}^k|s_h^k = s] = \mathbb{E}[\rho_{h:H+1}^k G_{h:H+1}^k|s_h^k = s]$.

We here explain why we set $\alpha = 1/H$. The main reason is that the performance of the estimates of $Q^\dagger$ depends on the number of the observations that follow $\pi^\dagger$. Even though IS estimator provides an unbiased estimate, the variance of the IS might be very high. By choosing $\alpha = 1/H$, we can control the variance. In particular, to obtain estimate $\hat{Q}_h^\dagger$, by the Bellman consistency equations, we first estimate $V_{h+1}^\dagger$. By setting $\alpha = \frac{1}{H}$, we can show that the importance ratio $\rho_{h+1:H+1}^k = \rho_{h+1:H}^k \leq \frac{1}{(1-\alpha)^{H-h}} \leq \frac{1}{(1-\alpha)^{H-1}} \leq e$ when $H \geq 2$ and $1 \leq h \leq H-1$, and $\rho_{H+1:H+1}^k = 1$. As the result, $\rho_{h+1:H}^k G_{h+1:H}^k$ will be bounded, and the variance of the estimate of $V_{h+1}^\dagger$ can be controlled.

We build a confidence bound to show the performance of the estimate error of $Q^\dagger$. The confidence bound is built based on Hoeffding inequalities and shown in the following Lemma.

**Lemma 2.** *If the attacker follows the LCB-H attack strategy on the RL agent, for any $p \in (0,1)$, with probability at least $1-p$, the following confidence bound of $\hat{Q}_h^\dagger$ holds simultaneously for all $(s,a,h,k) \in \mathcal{S} \times \mathcal{A} \times [H] \times [k]$:*

$$\left| \hat{Q}_{h,k}^\dagger(s,a) - Q_h^\dagger(s,a) \right| \leq (e(H-h)+1)\sqrt{2\log(2SAT/p)/N_h^k(s,a)}, \quad (11)$$

*where $\hat{Q}_{h,k}^\dagger(s,a)$ represents the attacker's evaluation of $Q$-values at the step $h$ at the beginning of the episode $k$, and $N_h^k(s,a)$ represents the cumulative number of attacker's state-action pair $(s,a)$ at the step $h$ until the beginning of the episode $k$, i.e. $N_h^k(s,a) = \sum_{k'=1}^{k-1} \mathbb{1}(s_h^{k'} = s)\mathbb{1}(\widetilde{a}_h^{k'} = a)$.*

The detailed proof can be found in Appendix C.1 of the supplementary material. In Lemma 2, the given bound on LCB-H attacker's estimation of the $Q$-values mainly based on $N_h^k(s,a)$ the number of state-action pair $(s,a)$ at the step $h$. This bound is similar to the confidence bound in the UCB algorithm for the bandit problem, except for the additional $H$ factor. Compared with the bandit problem, $Q$-values are the expected cumulative rewards, which bring the additional $H$ factor.

The LCB-H attack scheme uses a LCB method to explore and exploit the worst action. Thus, when the agent picks a non-target action, the LCB-H attacker changes it to different post-attack actions in different episodes. In the observation of the agent, the environment is non-stationary, i.e., the reward functions and probability transition function may change over episodes. Following the existing works on non-stationary RL [Fei et al., 2020, Mao et al., 2020, Cheung et al., 2020], we define $V_h^{k,\pi}(s) = \mathbb{E}\left[\sum_{h'=h}^H r_{h'}^k|s_h^k = s, \pi\right]$ and define the expected dynamic regret for the agent as:

$$\text{D-Regret}(K) = \sum_{k=1}^K [V_1^{k,\pi^{k,*}}(s_1^k) - V_1^{k,\pi^k}(s_1^k)], \quad (12)$$

where $\pi^{k,*}$ is the optimal policy at episode $k$, i.e. $V_h^{k,\pi^{k,*}}(s) = \sup_\pi V_h^{k,\pi}(s)$.

Here we state our main theorem, whose proof is deferred to the supplementary material.

**Theorem 2.** *Assume the expected dynamic regret of the RL agent's algorithm D-Regret$(K)$ is bounded by a sub-linear bound $\mathcal{R}(T)$, i.e., D-Regret$(K) \leq \mathcal{R}(T)$. With probability $1 - 4p$, the LCB-H attack will force the agent to learn the target policy $\pi^\dagger$ with the cost and loss bounded by*

$$\text{Cost}(K,H) \leq \text{Loss}(K,H) \leq \frac{H\left(\mathcal{R}(T) + 2H^2\sqrt{\log(1/p)\mathcal{R}(T)}\right)}{\Delta_{min}} + \frac{307SAH^4\log(2SAT/p)}{\Delta_{min}^2}.$$

From Theorem 2 we see that when $\mathcal{R}(T) \leq O(\frac{SAH^3\log(2SAT/p)}{\Delta_{min}})$, the cost and loss are bounded by $O(\frac{SAH^4\log(2SAT/p)}{\Delta_{min}^2})$, which scales as $\log(T)$, otherwise the cost and loss are bounded by $O(\mathcal{R}(T))$ that scales linearly with $\mathcal{R}(T)$. The LCB-H attack nearly matches the performance of the $\alpha$-portion attack, without requiring any information of the underlying environment and the agent's algorithm. Compared with the results of the $\alpha$-portion attack, the additional part of the bound in Theorem 2, i.e. $\frac{307SAH^4\log(2SAT/p)}{\Delta_{min}^2}$, is from the cost of exploring the worst action. Note that we use a LCB method to explore the worst action and the confidence bound is built by Hoeffding inequalities. The bound of the loss and cost can potentially be improved by using Bernstein-type concentration inequalities.

### 3.3 Black-box attack on UCB-H

In this section, we use UCB-H algorithm [Jin et al., 2018] as an example to illustrate the effects of the proposed LCB-H action poisoning attack strategy. UCB-H algorithm is a model-free $Q$-learning algorithm equipped with a UCB-Hoeffding exploration policy. At a high level, this algorithm builds an high-probability upper bound of $Q$-function for every state-action pairs. Then, it greedily chooses the action according to the optimistic estimations. In this section, we derive an upper bound of the loss and the cost of the LCB-H attack against UCB-H agent. Complete proofs of Theorem 3 and its supporting lemmas are provided in Appendix D of the supplementary material.

**Theorem 3.** *For any given target policy $\pi^\dagger \in \Pi^\dagger$, with probability $1 - 4p$, the LCB-H attacker can successfully manipulate the UCB-H algorithm to implement the target policy $\pi^\dagger$, with the cost and the loss bounded as follow:*

$$Cost(K,H) \leq Loss(K) \leq O\left(H^5\log(2H/p) + \frac{1}{\Delta_{min}}SAH^4 + \frac{1}{\Delta_{min}^2}H^{10}SA\log(2SAT/p)\right).$$

Theorem 3 reveals a significant security threat of efficient RL agents. It shows that by spending only logarithm cost, the LCB-H attack is able to force UCB-H agent to choose actions specified by a policy decided by the attack with only logarithm loss.

The results in Theorem 3 are consistent with the results in Theorem 2. In particular, [Yang et al., 2021] proved a gap-independence bound on UCB-H that scales as $O(\frac{H^6SA}{\Delta}\log(T))$, where $\Delta = \min_{h,s,a}\{V_h^*(s) - Q_h^*(s,a) : V_h^*(s) - Q_h^*(s,a) > 0\}$ is the sub-optimality gap. If an algorithm whose dynamic regret bound scales as $O(\frac{H^6SA}{\Delta}\log(T))$, the cost and loss are scale as $O(\frac{H^7SA}{\Delta^2}\log(T))$. UCB-H is a stationary RL algorithm, while the LCB-H adaptively attacks the agent and hence the effective environment observed by the agent is non-stationary. This adds a factor to the loss and cost.

## 4 Numerical Experiments

In this section, we empirically evaluate the performance of LCB-H attacks against three efficient RL agents, namely UCB-H [Jin et al., 2018], UCB-B [Jin et al., 2018] and UCBVI-CH [Azar et al., 2017], respectively. We perform numerical simulations on an environment represented as an MDP with ten states and five actions, i.e. $S = 10$ and $A = 5$. The environment is a periodic 1-d grid world. The action space $\mathcal{A}$ is given by {two steps left, one step left, stay, one step right, two steps right}. For any given state-action pair $(s,a)$, with probability $p(s,a)$, the agent navigates by the action; with probability $1 - p(s,a)$, the agent's next state is sampled randomly from the five adjacent states (include itself). For example, if the environment receives state-action pair $(s,a) = (5, \text{stay})$,

with probability $p(5, \text{stay})$, the next state is 5; with probability $\frac{1-p(5,\text{stay})}{5}$, the next state is 3, 4, 5, 6 or 7. By randomly generating $p(s, a)$ with $0.5 < p(s, a) < 1$, we randomly generate the transition probabilities $P(s'|s, a)$ for all action $a$ and state $s$. The mean reward of state-action pairs are randomly generated from a set of values $\{0.2, 0.35, 0.5, 0.65, 0.8\}$. In this paper, we assume the rewards are bounded by $[0, 1]$. Thus, we use Bernoulli distribution to randomize the reward signal. The target policy is randomly chosen by deleting the worst action, so as to satisfy Assumption 1. We set the total number of steps $H = 10$ and the total number of episodes $K = 10^9$.

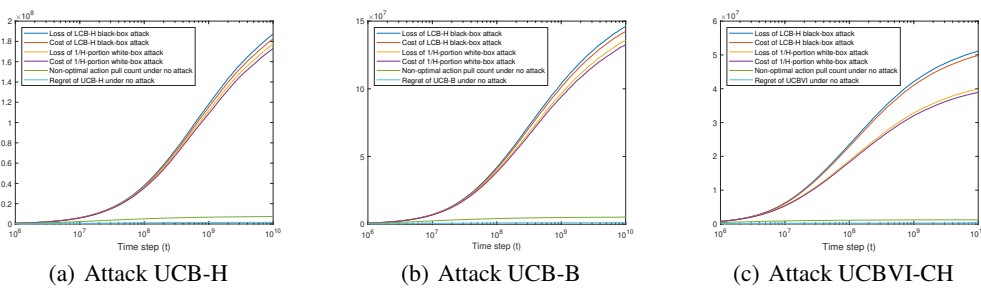

| (a) Attack UCB-H | (b) Attack UCB-B | (c) Attack UCBVI-CH |

Figure 1: Action poisoning attacks against RL agents

In Figure 1, we illustrate $\frac{1}{H}$-portion white-box attack and LCB-H black-box attack against three different agents separately and compare the loss and cost of these two attack schemes. For comparison purposes, we also add the curves for the regret of three agents under no attack. In the figure, the non-optimal action pull count are defined as $\sum_{k=1}^{K} \sum_{h=1}^{H} \mathbb{1}(Q_h^*(s_h^k, a_h^k) < V_h^*(s_h^k))$. The x-axis uses a base-10 logarithmic scale and represents the time step $t$ with the total time step $T = KH$. The y-axis represents the cumulative loss, cost and regret that change over time steps. The results show that, the loss and cost of $\frac{1}{H}$-portion white-box attack and LCB-H black-box attack scale as $\log(T)$. Furthermore the performances of our black-box attack scheme, LCB-H, nearly matches those of the $\frac{1}{H}$-portion white-box attack scheme. In addition, the cost and loss are about $H/\Delta_{min}$ times as much as the regret. This is consistent with our analysis in Theorem 2. Each of the individual experimental runs costs about twenty hours on one physical CPU core. The type of CPU is Intel Core i7-8700. More numerical results can be found in the appendix.

## 5 Limitations

Here we highlight the assumptions and limitations of our work. Our theoretical results rely on Assumption 1 which limits the choice of the target policy. A violation of Assumption 1 may cause linear cost or loss of the proposed attack scheme. In Theorem 2, we assume the expected dynamic regret of the RL agent is bounded. Generally, the expected dynamic regret is a stronger notation than the dynamic regret. In other words, the optimal policy $\pi^{k,*}$ in (12) may change in each episode $k$, while the optimal policy $\pi^k$ in (1) is fixed over episodes. In this paper, we discuss the action poisoning attack in the tabular episodic MDP context. Although we are convinced that the idea of our proposed attack scheme can be carried over to RL with function approximation, the current results only apply to the tabular episodic MDP setting.

## 6 Conclusions and Discussion

In this paper, we have introduced a new class of attacks on RL: action poisoning attacks. We have proposed the $\alpha$-portion white-box attack and the LCB-H black-box attack. We have shown that the $\alpha$-portion white-box attack is able to attack any efficient RL agent and the LCB-H black-box attack nearly matches the performance of the $\alpha$-portion attack. We have analyzed the LCB-H attack against the UCB-H algorithm and proved that the proposed attack scheme can force the agent to almost always follow a particular class of target policy with only logarithm loss and cost. In the future, we will investigate action poisoning attacks on other RL models such as multi-agent RL model. It is also of interest to investigate the defense strategy to mitigate the effects of this attack.

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
