# A Additional numerical experiments

In this section, we introduce some additional numerical experiments. We perform numerical simulations on an environment represented as an MDP with 12 states and 4 actions, i.e. $S = 12$ and $A = 4$. The environment is a 4-by-4 grid world. The action space $\mathcal{A}$ is given by {North = 1, South = 2, West = 3, East = 4}. The terminal state is at cell $[4, 4]$ (blue cell). If the agent at the terminal state and chooses any actions, the next state will be the beginning state at cell $[1, 1]$ and the agent receives reward $+1$. The agent is blocked by obstacles in cells $[2, 2], [2, 3], [2, 4]$ and $[3, 2]$ (black cells). The environment contains a special jump from cell $[1, 3]$ to cell $[3, 3]$ with $+1$ reward. When the agent at the cell $[1, 3]$ and chooses action "South", the agent will jump to the cell $[3, 3]$. Actions that would take the agent off the grid leave its location unchanged.

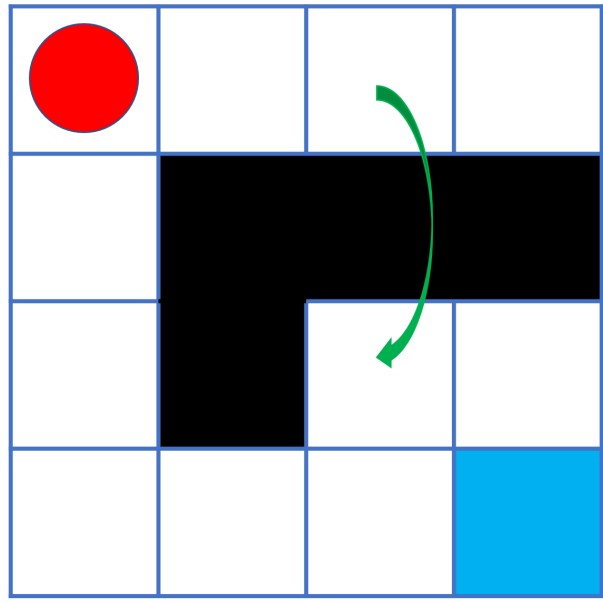

Figure 2: 2-d grid world

To add some randomness of the environment, we set that the states transit randomly. After the environment receives the action signal, the next state may generated by following any of the other three actions with probability 0.1 separately. For example, if the agent at cell $[4, 3]$ and chooses action "North", the next state will be $[3, 3]$ with probability 0.7, $[4, 2]$ with probability 0.1, $[4, 3]$ with probability 0.1, or $[4, 4]$ with probability 0.1. The mean rewards of actions that would take the agent off the grid or towards the obstacle are 0. The mean rewards of other state-action pairs are 0.2 or 0.4. In this paper, we assume the rewards are bounded by $[0, 1]$. Thus, we use Bernoulli distribution to randomize the reward signal. The optimal policy encourages the agent to take the special jump and reach the terminal state. In the target policy, the agent will reach the terminal state as soon as possible but avoid to take the special jump. We set the total number of steps $H = 10$ and the total number of episodes $K = 10^9$. We empirically evaluate the performance of LCB-H attacks against three efficient RL agents, namely UCB-H [Jin et al., 2018], UCB-B [Jin et al., 2018] and UCBVI-CH [Azar et al., 2017], respectively.

In Figure 3, we illustrate $\frac{1}{H}$-portion white-box attack and LCB-H black-box attack against three different agents separately and compare the loss and cost of these two attack schemes. For comparison purposes, we also add the curves for the regret of three agents under no attack. The x-axis uses a base-10 logarithmic scale and represents the time step $t$ with the total time step $T = KH$. As same as the results in Figure 1 the results show that the loss and cost of $\frac{1}{H}$-portion white-box attack and LCB-H black-box attack scale as $\log(T)$. Furthermore the performances of our black-box attack scheme, LCB-H, nearly matches those of the $\frac{1}{H}$-portion white-box attack scheme.

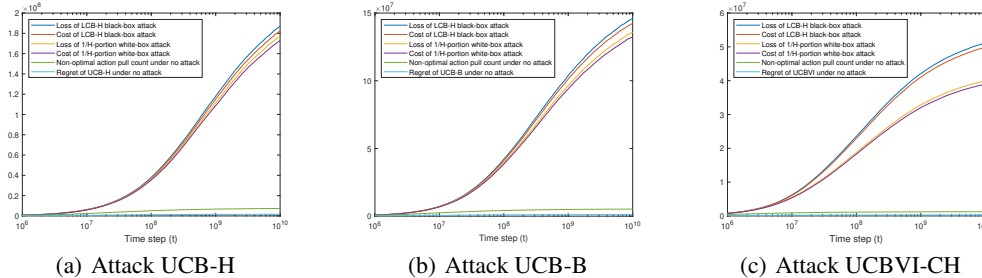

|            |            |               |
| :--------: | :--------: | :-----------: |
| (a) Attack UCB-H | (b) Attack UCB-B | (c) Attack UCBVI-CH |

Figure 3: Action poisoning attacks against RL agents

## B  Proofs for the white-box attack

### B.1  Proof of Lemma 1

We assume that the agent does not know the attacker's manipulations and the presence of the attacker. We can consider the combination of the attack and the environment as a new environment, and the RL agent interacts with the new environment in the attack setting. We define $\overline{Q}$ and $\overline{V}$ as the $Q$-values and value functions of the new environment that the RL agent observes. The optimal policy can be given from the the Bellman optimality equations. Suppose the target policy $\pi^\dagger$ is optimal at step $h+1$ in the observation of the agent. Then, $\overline{V}^*_{h+1}(s) = \overline{V}^\dagger_{h+1}(s)$ for all state $s$, where $\overline{V}$ represents the value function in the observation of the agent. Similarly, we set $\overline{Q}$ as the $Q$-values in the observation of the agent. As the attacker does not attack when the agent pick the target action, $\overline{V}^\dagger_{h+1} = V^\dagger_{h+1}$. For any $a \neq \pi^\dagger_h(s)$, from the equation (3), (4) and (6), $\overline{Q}^*_h$ is given by

$$
\begin{aligned}
\overline{Q}^*_h(s,a) =& (1-\alpha)(R_h(s,\pi^\dagger_h(s)) + P_h\overline{V}^*_{h+1}(s,\pi^\dagger_h(s))) \\
& + \alpha(R_h(s,\pi^-_h(s)) + P_h\overline{V}^*_{h+1}(s,\pi^-_h(s))) \\
=& (1-\alpha)Q^\dagger_h(s,\pi^\dagger_h(s)) + \alpha Q^\dagger_h(s,\pi^-_h(s)) \\
<& Q^\dagger_h(s,\pi^\dagger_h(s)) = V^\dagger_h(s) = \overline{Q}^\dagger_h(s,\pi^\dagger_h(s)).
\end{aligned}
\tag{13}
$$

We can conclude that if the target policy $\pi^\dagger$ is optimal at step $h+1$ in the observation of the agent, the target policy $\pi^\dagger$ is also optimal at step $h$ in the observation of the agent. Since $V^\pi_{H+1} = \mathbf{0}$ and $Q^\pi_{H+1} = \mathbf{0}$, the target policy $\pi^\dagger$ is the optimal policy, from induction on $h = H, H-1, \cdots, 1$.

### B.2  Proof of Theorem 1

Here, we follows the idea of error decomposition proposed in [Yang et al., 2021, He et al., 2020]. We first decomposed the expected regret Regret$(K)$ into the gap of $Q$-values. Denote by $\Delta^k_h = V^\dagger_h(s^k_h) - \min_{a\in\mathcal{A}} Q^\dagger_h(s^k_h,a)$ and $\overline{\Delta}^k_h = \overline{V}^\dagger_h(s^k_h) - \overline{Q}^\dagger_h(s^k_h,a^k_h)$.

As shown in Lemma 1, the target policy $\pi^\dagger$ is optimal in the observation of the agent. Thus,

$$
\text{Regret}(K) = \sum_{k=1}^K [\overline{V}^*_1(s^k_1) - \overline{V}^{\pi^k}_1(s^k_1)] = \sum_{k=1}^K [\overline{V}^\dagger_1(s^k_1) - \overline{V}^{\pi^k}_1(s^k_1)].
\tag{14}
$$

For episode $k$,

$$\overline{V}_1^\dagger(s_1^k) - \overline{V}_1^{\pi^k}(s_1^k)$$

$$=\overline{V}_1^\dagger(s_1^k) - \mathbb{E}_{a\sim\pi_1^k(\cdot|s_1^k)}[\overline{Q}_1^\dagger(s_1^k,a)|\mathcal{F}_1^k] + \mathbb{E}_{a\sim\pi_1^k(\cdot|s_1^k)}[\overline{Q}_1^\dagger(s_1^k,a)|\mathcal{F}_1^k] - \overline{V}_1^{\pi^k}(s_1^k)$$

$$=\mathbb{E}[\overline{\Delta}_1^k|\mathcal{F}_1^k] + \mathbb{E}_{s'\sim P_1(\cdot|s_1^k,a\sim\pi_1^k(\cdot|s_1^k))}[(\overline{V}_2^\dagger - \overline{V}_2^{\pi^k})(s')]$$

$$=\cdots = \mathbb{E}[\sum_{h=1}^H \overline{\Delta}_h^k|\mathcal{F}_1^k] \tag{15}$$

$$\overset{\textcircled{1}}{=}\mathbb{E}[\sum_{h=1}^H \alpha\Delta_h^k \mathbb{1}(a_h^k \neq \pi_h^\dagger(s_h^k))|\mathcal{F}_1^k]$$

$$\geq\alpha\Delta_{min}\mathbb{E}[\sum_{h=1}^H \mathbb{1}(\widetilde{a}_h^k \neq a_h^k)],$$

where $\mathcal{F}_h^k$ represents the $\sigma$-field generated by all the random variables until episode $k$, step $h$ begins, and the equation ① holds due to $\overline{Q}_h^\dagger(s_h^k,a_h^k) = (1-\alpha)Q_h^\dagger(s_h^k,\pi_h^\dagger(s_h^k)) + \alpha Q_h^\dagger(s_h^k,\pi_h^-(s_h^k))$ when $a_h^k \neq \pi_h^\dagger(s_h^k)$, and $\overline{V}_h^\dagger(s_h^k) = V_h^\dagger(s_h^k)$.

In the $\alpha$-portion attack, the attacker attacks only when the agent picks a non-target arm. Thus, $\mathbb{1}(\widetilde{a}_h^k \neq a_h^k) \leq \mathbb{1}(a_h^k \neq \pi_h^\dagger(s_h^k))$ and $\mathrm{Cost}(K,H) \leq \mathrm{Loss}(K,H)$.

We can conclude that

$$\mathbb{E}[\mathrm{Cost}(K,H)] \leq \mathbb{E}[\mathrm{Loss}(K,H)] \leq \frac{\mathrm{Regret}(K)}{\alpha\Delta_{min}}. \tag{16}$$

Before the proof of the upper bound on the loss and the cost, we first introduce an important lemma, which shows the connections between the expected regret to the loss and the cost.

**Lemma 3.** *For any MDP $\mathcal{M} = (\mathcal{S},\mathcal{A},H,P,R)$ and any $p \in (0,1)$, with probability at least $1-p$, we have*

$$\sum_{k=1}^K\sum_{h=1}^H \overline{\Delta}_h^k \leq \sum_{k=1}^K \left(\overline{V}_h^\dagger(s_h^k) - \overline{V}_h^{\pi^k}(s_h^k)\right) + 2H^2\sqrt{\log(1/p)\sum_{k=1}^K\left(\overline{V}_h^\dagger(s_h^k) - \overline{V}_h^{\pi^k}(s_h^k)\right)}. \tag{17}$$

The proof of Lemma 3 is based on the Freedman inequality [Freedman, 1975, Tropp et al., 2011]. Since $\mathbb{E}[\sum_{h=1}^H \overline{\Delta}_h^k|\mathcal{F}_1^k] = \overline{V}_1^\dagger(s_1^k) - \overline{V}_1^{\pi^k}(s_1^k)$, denote by $X_k = \sum_{h=1}^H \overline{\Delta}_h^k - \left(\overline{V}_h^\dagger(s_h^k) - \overline{V}_h^{\pi^k}(s_h^k)\right)$, then $\{X_k\}_{k=1}^K$ is a martingale difference sequence w.r.t the filtration $\{\mathcal{F}_1^k\}_{k\geq1}$. The difference sequence is uniformly bounded by $|X_k^2| \leq H^2$. Define the predictable quadratic variation process of the martingale $W_K := \sum_{k=1}^K \mathbb{E}[X_k^2|\mathcal{F}_1^k]$, which is bounded by

$$W_K \leq \sum_{k=1}^K \mathbb{E}\left[\left(\overline{\Delta}_h^k\right)^2|\mathcal{F}_1^k\right] \leq \sum_{k=1}^K H^2\mathbb{E}\left[\overline{\Delta}_h^k|\mathcal{F}_1^k\right] = \sum_{k=1}^K H^2\left(\overline{V}_h^\dagger(s_h^k) - \overline{V}_h^{\pi^k}(s_h^k)\right). \tag{18}$$

By the Freedman's inequality, we have

$$\mathbb{P}\left(\sum_{k=1}^K X_k > 2H^2\sqrt{\log(1/p)\sum_{k=1}^K\left(\overline{V}_h^\dagger(s_h^k) - \overline{V}_h^{\pi^k}(s_h^k)\right)}\right)$$

$$\leq\exp\left\{\frac{-2H^4\log(1/p)\sum_{k=1}^K\left(\overline{V}_h^\dagger(s_h^k) - \overline{V}_h^{\pi^k}(s_h^k)\right)}{W_K + H^2*2H^2\sqrt{\log(1/p)\sum_{k=1}^K\left(\overline{V}_h^\dagger(s_h^k) - \overline{V}_h^{\pi^k}(s_h^k)\right)}/3}\right\} \tag{19}$$

$$\leq\exp\left\{-\log(1/p)\right\} = p.$$

Theorem 1 is directly from Lemma 3 and $\overline{\Delta}_h^k \geq \alpha \Delta_{min} \mathbb{1}(\pi_h^\dagger(s_h^k) \neq \pi_h^k(s_h^k))$.

## C Proofs for LCB-H attack

### C.1 Proof of Lemma 2

At the beginning of the episode $k$, for any step $h \in [H]$ and any $(s,a) \in \mathcal{S} \times \mathcal{A}$ with $N_h^k(s,a) \neq 0$, according to Algorithm 1, the estimate of $Q$-values under the target policy $\pi^\dagger$ are given by

$$\hat{Q}_{h,k}^\dagger(s,a) = \frac{1}{N_h^k(s,a)} \sum_{i=1}^{k-1} \mathbb{1}\left((\tilde{a}_h^k, s_h^k) = (s,a)\right) \left(r_h^k + \rho_{h+1:H+1}^k G_{h+1:H+1}^k\right). \qquad (20)$$

Note that for any $\left((\tilde{a}_h^k, s_h^k) = (s,a)\right)$, we have

$$\mathbb{E}[r_h^k + \rho_{h+1:H+1}^k G_{h+1:H+1}^k | \tilde{a}_h^k, s_h^k] = R_h(s,a) + \mathbb{E}_{s' \sim P_h(\cdot|s,a)}[V_{h+1}^\dagger(s')] = Q_h^\dagger(s,a). \qquad (21)$$

Thus, we can apply Hoeffding's inequality here to bound $|\hat{Q}_{h,k}^\dagger(s,a) - Q_h^\dagger(s,a)|$. The cumulative reward is bounded by $0 \leq G_{h+1:H+1}^k \leq H - h$ and the important sampling ratio is bounded by $0 \leq \rho_{h+1:H+1}^k \leq e$ because

$$\rho_{h+1:H+1}^k \leq \left(\frac{1}{(1-\frac{1}{H})}\right)^{H-h} \leq \left(\frac{1}{(1-\frac{1}{H})}\right)^{H-1} \leq e. \qquad (22)$$

By Hoeffding's inequality, since $|r_h^k + \rho_{h+1:H+1}^k G_{h+1:H+1}^k| \leq e(H-h) + 1$, we have

$$\mathbb{P}\left(|\hat{Q}_{h,k}^\dagger(s,a) - Q_h^\dagger(s,a)| > \eta\right) \leq 2\exp\left(-\frac{\eta^2}{2N_h^k(s,a)\left(\frac{H-h+1}{N_h^k(s,a)}\right)^2}\right). \qquad (23)$$

To hold a high-probability confidence bound for any state $s$, any action $a$, any step $h$ and any episode $k$, set the right hand side of the above inequality to $p/SAT$. Then, we have $\eta = (e(H-h)+1)\sqrt{\frac{2\iota}{N_h^k(s,a)}}$ and $\iota = \log(2SAT/p)$.

### C.2 Proof of Theorem 2

From Lemma 3, for any MDP $\mathcal{M} = (\mathcal{S}, \mathcal{A}, H, P, R)$ and any $p \in (0,1)$, with probability at least $1-p$, we have

$$\sum_{k=1}^K \sum_{h=1}^H \overline{\Delta}_h^k \leq \sum_{k=1}^K \left(\overline{V}_h^{k,\dagger}(s_h^k) - \overline{V}_h^{k,\pi^k}(s_h^k)\right) + 2H^2 \sqrt{\log(1/p) \sum_{k=1}^K \left(\overline{V}_h^\dagger(s_h^k) - \overline{V}_h^{k,\pi^k}(s_h^k)\right)}$$

$$\leq \sum_{k=1}^K \left(\overline{V}_h^{k,*}(s_h^k) - \overline{V}_h^{k,\pi^k}(s_h^k)\right) + 2H^2 \sqrt{\log(1/p) \sum_{k=1}^K \left(\overline{V}_h^{k,*}(s_h^k) - \overline{V}_h^{k,\pi^k}(s_h^k)\right)}$$

$$= \text{D-Regret}(K) + 2H^2\sqrt{\log(1/p)\text{D-Regret}(K)}. \qquad (24)$$

Since the LCB-H attacker dose not attack the target action, $\overline{V}_h^{k,\dagger}(s_h^k) = V_h^\dagger(s_h^k)$. Thus, we have $\overline{\Delta}_h^k = \overline{V}_h^{k,\dagger}(s_h^k) - \overline{Q}_h^{k,\dagger}(s_h^k, a_h^k) = V_h^\dagger(s_h^k) - \overline{Q}_h^{k,\dagger}(s_h^k, a_h^k)$. When the agent picks a target action $a_h^k = \pi^\dagger(s_h^k)$, the attacker does not attack and $\overline{Q}_h^{k,\dagger}(s_h^k, a_h^k) = \overline{V}_h^{k,\dagger}(s_h^k) = V_h^\dagger(s_h^k)$. Thus, the left hand side of the equation (24) can be written as

$$\sum_{k=1}^K \sum_{h=1}^H \overline{\Delta}_h^k = \sum_{k=1}^K \sum_{h=1}^H \mathbb{1}(a_h^k \neq \pi_h^\dagger(s_h^k))\overline{\Delta}_h^k = \sum_{(k,h)\in\tau} \overline{\Delta}_h^k, \qquad (25)$$

where $\tau = \{(k, h) \in [K] \times [H] | a_h^k \neq \pi_h^\dagger(s_h^k)\}$.

At episode $k$ and step $h$, after the agent picks an action, since the attack scheme is given, we have $\overline{Q}_h^{k,\dagger}(s_h^k, a_h^k) = \mathbb{E}[Q_h^\dagger(s_h^k, \widetilde{a}_h^k)|\mathcal{F}_1^k, s_h^k, a_h^k]$. Furthermore, $\mathbb{E}[V_h^\dagger(s_h^k) - Q_h^\dagger(s_h^k, \widetilde{a}_h^k)|\mathcal{F}_1^k, s_h^k, a_h^k] = \overline{\Delta}_h^k$. By the Hoeffding inequality, since $|V_h^\dagger(s_h^k) - Q_h^\dagger(s_h^k, \widetilde{a}_h^k)| \leq H$, we have

$$\mathbb{P}\left(\sum_{(k,h) \in \tau} \left(V_h^\dagger(s_h^k) - Q_h^\dagger(s_h^k, \widetilde{a}_h^k) - \overline{\Delta}_h^k\right) > \eta\right) \leq \exp\left(-\frac{\eta^2}{2|\tau|H^2}\right). \tag{26}$$

Set the left hand side of the above inequality to $p$. With probability $1 - p$, we have,

$$\sum_{k=1}^K \sum_{h=1}^H \overline{\Delta}_h^k \geq \sum_{(k,h) \in \tau} \left(V_h^\dagger(s_h^k) - Q_h^\dagger(s_h^k, \widetilde{a}_h^k)\right) - H\sqrt{2|\tau|\log(1/p)}. \tag{27}$$

If $\widetilde{a}_h^k \neq \pi_h^\dagger(s)$ holds, the attacker attacked the agent, and from Lemma 2, we have with probability $1 - p$,

$$Q_h^\dagger(s, \pi_h^-(s)) \geq L_h^k(s, \pi_h^-(s)) \geq L_h^k(s, \widetilde{a}_h^k) \geq Q_h^\dagger(s, \widetilde{a}_h^k) - 2(e(H-h)+1)\sqrt{\frac{2\iota}{N_h^k(s_h^k, \widetilde{a}_h^k)}}, \tag{28}$$

and $0 \leq Q_h^\dagger(s, \widetilde{a}_h^k) - Q_h^\dagger(s, \pi_h^-(s)) \leq 2(e(H-h)+1)\sqrt{\frac{2\iota}{N_h^k(s_h^k, \widetilde{a}_h^k)}}$. If $\widetilde{a}_h^k \neq \pi_h^\dagger(s)$ holds, $V_h^\dagger(s_h^k) = Q_h^\dagger(s_h^k, \widetilde{a}_h^k)$. For the second item in the right hand side of inequality (27), we have with probability $1 - p$,

$$\sum_{(k,h) \in \tau} \left(V_h^\dagger(s_h^k) - Q_h^\dagger(s_h^k, \widetilde{a}_h^k)\right)$$
$$\geq \sum_{(k,h) \in \tau} \mathbb{1}\left(\widetilde{a}_h^k \neq \pi_h^\dagger(s)\right)\left(\Delta_h^k - 2(e(H-h)+1)\sqrt{\frac{2\iota}{N_h^k(s_h^k, \widetilde{a}_h^k)}}\right). \tag{29}$$

For $(k, h) \in \tau$, $\mathbb{E}[\mathbb{1}(\widetilde{a}_h^k \neq \pi_h^\dagger(s))|\mathcal{F}_h^k, (k, h) \in \tau] = 1/H$. By the Hoeffding inequality, we have with probability $1 - p$,

$$\sum_{(k,h) \in \tau} \left|\mathbb{1}\left(\widetilde{a}_h^k \neq \pi_h^\dagger(s)\right) - 1/H\right| \leq \sqrt{2|\tau|log(2/p)}. \tag{30}$$

We regroup the right hand side of inequality (29) in a different way and further

$$\sum_{(k,h)\in\tau} \mathbb{1}\left(\widetilde{a}_h^k \neq \pi_h^\dagger(s)\right) \sqrt{1/N_h^k(s_h^k, \widetilde{a}_h^k)}$$

$$= \sum_{h\in[H]} \sum_{s\in\mathcal{S}} \sum_{a\neq\pi_h^\dagger(s)} \sum_{n=1}^{N_h^{K+1}(s,a)} \sqrt{1/n}$$

$$\leq \sum_{h\in[H]} \sum_{s\in\mathcal{S}} \sum_{a\neq\pi_h^\dagger(s)} \left(1 + \int_{n=1}^{N_h^{K+1}(s,a)} \sqrt{1/n}\, dn\right)$$

$$\leq \sum_{h\in[H]} \sum_{s\in\mathcal{S}} \sum_{a\neq\pi_h^\dagger(s)} 2\sqrt{N_h^{K+1}(s,a)} \tag{31}$$

$$\overset{\textcircled{1}}{\leq} 2SAH\sqrt{\frac{\sum_{h\in[H]} \sum_{s\in\mathcal{S}} \sum_{a\neq\pi_h^\dagger(s)} N_h^{K+1}(s,a)}{SAH}}$$

$$= 2\sqrt{SAH \sum_{(k,h)\in\tau} \mathbb{1}\left(\widetilde{a}_h^k \neq \pi_h^\dagger(s)\right)}$$

$$\overset{\textcircled{2}}{\leq} 2\sqrt{SAH\left(|\tau|/H + \sqrt{2|\tau|log(2/p)}\right)}$$

$$\leq 2\sqrt{SA|\tau|} + 2\sqrt{2SAH|\tau|log(2/p)},$$

where ① holds due to the property of the concave function $\sqrt{n}$ and ② holds due to the inequality (30). In addition,

$$\sum_{(k,h)\in\tau} \mathbb{1}\left(\widetilde{a}_h^k \neq \pi_h^\dagger(s)\right) \Delta_h^k \geq \left(|\tau|/H - \sqrt{2|\tau|log(2/p)}\right) \Delta_{min}. \tag{32}$$

Combing (24), (27), (29), (31) and (32), we have

$$\Delta_{min}|\tau|/H \leq \text{D-Regret}(K) + 2H^2\sqrt{\log(1/p)\text{D-Regret}(K)} + (H + \Delta_{min})\sqrt{2|\tau|\log(1/p)}$$
$$2(e(H-h)+1)\sqrt{2\iota}\left(2\sqrt{SA|\tau|} + 2\sqrt{2SAH|\tau|log(2/p)}\right), \tag{33}$$

which is equivalent to

$$|\tau| \leq \frac{H}{\Delta_{min}}\left(\text{D-Regret}(K) + 2H^2\sqrt{\log(1/p)\text{D-Regret}(K)}\right) + \frac{307SAH^4\iota}{\Delta_{min}^2}. \tag{34}$$

In addition, $\text{Cost}(K,H) \leq \text{Loss}(K,H) = \sum_{k=1}^{K} \sum_{h=1}^{H} \mathbb{1}(a_h^k \neq \pi_h^\dagger(s_h^k)) = |\tau|$. The proof is completed.

## D   Proof of LCB-H attacks on UCB-H

For completeness, we describe the main steps of UCB-H algorithm in Algorithm 2.

Before the proof of Theorem 3, we first introduce our main technical lemma.

We denote by $\overline{Q}_h^k$, $\overline{V}_h^k$, $\overline{N}_h^k$ the observations of UCB-H agent at the beginning of episode $k$. The lemma below is our main technical lemma that shows the difference between the agent's observations $\overline{Q}_h^k$ and the true $Q$-values $Q_h^\dagger$ can be bounded by quantities from the next step.

**Lemma 4.** *Assume the attacker follows the LCB-H attack strategy on the UCB-H agent. Suppose the constant $c$ in UCB-H algorithm satisfies $c > 0$. Let $\beta_h(t) = (cH + 2(H-h) + 2)\sqrt{H\iota/t}$ when $t > 0$ and $\beta_h(0) = 0$ for any step $h$, and let $B_h(t) = (e(H-h)+1)\sqrt{\frac{2\iota}{t}}$ when $t > 0$*

**Algorithm 2:** Q-learning with UCB-Hoeffding [Jin et al., 2018]

---

1: Initialize $Q_h(s,a) = 0$ and $N_h(s,a) = 0$ for all state $s \in \mathcal{S}$, all action $a \in \mathcal{A}$ and all step $h \in [H]$.
2: Define $\alpha_t = \frac{H+1}{H+t}$, $\iota = log(2SAT/p)$, and set a constant $c$.
3: **for** episode $k = 1, 2, \ldots, K$ **do**
4:    Receive $s_1$.
5:    **for** step $h = 1, 2, \ldots, H$ **do**
6:       Take action $a_h \leftarrow \arg\max_{a'} Q_h(s_h, a')$, and observe $s_{h+1}$ and $r_h$.
7:       $t = N_h(s_h, a_h) \leftarrow N_h(s_h, a_h) + 1$; $b_t = c\sqrt{H^3\iota/t}$.
8:       $Q_h(s_h, a_h) = (1 - \alpha_t)Q_h(s_h, a_h) + \alpha_t[r_h + V_{h+1}(s_{h+1}) + b_t]$.
9:       $V_h(s_h) \leftarrow \min\{H, \max_{a'} Q_h(s_h, a')\}$.
10:    **end for**
11: **end for**

---

and $B_h(0) = H$ for any step $h$. For any $p \in (0,1)$, with probability at least $1 - 3p$, the following confidence bounds hold simultaneously for all $(s, a, h, k) \in \mathcal{S} \times \mathcal{A} \times [H] \times [k]$:

$$\sum_{i=1}^{t} \alpha_t^i \left( \overline{V}_{h+1}^{k_i}(s_{h+1}^{k_i}) - V_{h+1}^\dagger(s_{h+1}^{k_i}) \right) \leq \overline{Q}_h^k(s, \pi_h^\dagger(s)) - Q_h^\dagger(s, \pi_h^\dagger(s))$$

$$\leq \alpha_t^0 H + \sum_{i=1}^{t} \alpha_t^i \left( \overline{V}_{h+1}^{k_i}(s_{h+1}^{k_i}) - V_{h+1}^\dagger(s_{h+1}^{k_i}) \right) + \beta_h(t), \tag{35}$$

*and*

$$\overline{Q}_h^k(s, a) - Q_h^\dagger(s, \pi_h^\dagger(s)) = \overline{Q}_h^k(s, a) - Q_h^\dagger(s, \pi_h^-(s)) - \Delta_h(s)$$

$$\leq \alpha_t^0 H + \sum_{i=1}^{t} \alpha_t^i \left( \overline{V}_{h+1}^{k_i}(s_{h+1}^{k_i}) - V_{h+1}^\dagger(s_{h+1}^{k_i}) \right) + \beta_h(t)$$

$$+ \sum_{i=1}^{t} \alpha_t^i \mathbb{1}\left( \widetilde{a}_h^{k_i} \neq \pi_h^\dagger(s) \right) \left( 2B_h\left( N_h^{k_i}(s, \widetilde{a}_h^{k_i}) \right) - \Delta_h(s) \right), \tag{36}$$

where $t = \overline{N}_h^k(s, a)$, $\Delta_h(s) := Q_h^\dagger(s, \pi_h^\dagger(s)) - Q_h^\dagger(s, \pi_h^-(s))$, and $k_1, k_2, \ldots, k_t < k$ are the episodes in which $(s, a)$ was previously taken by the agent at step $h$.

By recursing the results in Lemma 4, we can obtained Theorem 3.

### D.1 Proof of Lemma 4

Lemma 4 shows the result of the LCB-H attacks on the UCB-H algorithm. Thus, we need to refer the readers to some settings and the Lemma 4.1 in [Jin et al., 2018]. Note that UCB-H chooses the learning rate as $\alpha_t := \frac{H+1}{H+t}$. For notational convenience, define $\alpha_t^0 := \prod_{j=1}^{t}(1 - \alpha_t)$ and $\alpha_t^i := \alpha_i \prod_{j=i+1}^{t}(1 - \alpha_t)$. Here, we introduce some useful properties of $\alpha_t^i$ which were proved in [Jin et al., 2018]:
(1) $\sum_{i=1}^{t} \alpha_t^i = 1$ and $\alpha_t^0 = 0$ for $t \geq 1$;
(2) $\sum_{i=1}^{t} \alpha_t^i = 0$ and $\alpha_t^0 = 1$ for $t = 0$;
(3) $\frac{1}{\sqrt{t}} \leq \sum_{i=1}^{t} \frac{\alpha_t^i}{\sqrt{t}} \leq \frac{2}{\sqrt{t}}$ for every $t \geq 1$;
(4) $\sum_{i=1}^{t} (\alpha_t^i)^2 \leq \frac{2H}{t}$ for every $t \geq 1$;
(5) $\sum_{t=i}^{\infty} \alpha_t^i \leq (1 + \frac{1}{H})$ for every $i \geq 1$.

As shown in [Jin et al., 2018], at any $(s, a, h, k) \in \mathcal{S} \times \mathcal{A} \times [H] \times [K]$, let $t = \overline{N}_h^k(s, a)$ and suppose $(s, a)$ was previously taken by the agent at step $h$ of episodes $k_1, k_2, \ldots, k_t < k$. By the update equations in the UCB-H Algorithm and the definition of $\alpha_t^i$, we have

$$\overline{Q}_h^k(s, a) = \alpha_t^0 H + \sum_{i=1}^{t} \alpha_t^i \left( r_h^{k_i} + \overline{V}_{h+1}^{k_i}(s_{h+1}^{k_i}) + b_i \right). \tag{37}$$

Then we can bound the difference between $\overline{Q}_h^k$ and $Q_h^\dagger$.

$$\overline{Q}_h^k(s,a) - Q_h^\dagger(s, \pi_h^-(s))$$
$$=\alpha_t^0 \left( H - Q_h^\dagger(s, \pi_h^-(s)) \right)$$
$$+ \sum_{i=1}^t \alpha_t^i \left( r_h^{k_i} + \overline{V}_{h+1}^{k_i}(s_{h+1}^{k_i}) + b_i - Q_h^\dagger(s, \pi_h^-(s)) \right)$$
$$=\alpha_t^0 (H - Q_h^\dagger(s, \pi_h^-(s)) + \sum_{i=1}^t \alpha_t^i \left( r_h^{k_i} - r_h(s, \widetilde{a}_h^{k_i}) + b_i \right)$$
$$+ \sum_{i=1}^t \alpha_t^i \left( r_h(s, \widetilde{a}_h^{k_i}) + \overline{V}_{h+1}^{k_i}(s_{h+1}^{k_i}). - Q_h^\dagger(s, \pi_h^-(s)) \right). \tag{38}$$

We can rewrite the third term in the RHS of (38) as follows

$$r_h(s, \widetilde{a}_h^{k_i}) + \overline{V}_{h+1}^{k_i}(s_{h+1}^{k_i}) - Q_h^\dagger(s, \pi_h^-(s))$$
$$=r_h(s, \widetilde{a}_h^{k_i}) + \overline{V}_{h+1}^{k_i}(s_{h+1}^{k_i}) - Q_h^\dagger(s, \widetilde{a}_h^{k_i}) + Q_h^\dagger(s, \widetilde{a}_h^{k_i}) - Q_h^\dagger(s, \pi_h^-(s))$$
$$=\overline{V}_{h+1}^{k_i}(s_{h+1}^{k_i}) - P_h V_{h+1}^\dagger(s, \widetilde{a}_h^{k_i}) + Q_h^\dagger(s, \widetilde{a}_h^{k_i}) - Q_h^\dagger(s, \pi_h^-(s)) \tag{39}$$
$$=\overline{V}_{h+1}^{k_i}(s_{h+1}^{k_i}) - V_{h+1}^\dagger(s_{h+1}^{k_i}) + V_{h+1}^\dagger(s_{h+1}^{k_i}) - P_h V_{h+1}^\dagger(s, \widetilde{a}_h^{k_i})$$
$$+ Q_h^\dagger(s, \widetilde{a}_h^{k_i}) - Q_h^\dagger(s, \pi_h^-(s)).$$

As the result, the difference between $\overline{Q}_h^k$ and $Q_h^\dagger$ can be rewritten as

$$\overline{Q}_h^k(s,a) - Q_h^\dagger(s, \pi_h^-(s))$$
$$=\alpha_t^0(H - Q_h^\dagger(s, \pi_h^-(s)) + \sum_{i=1}^t \alpha_t^i \left( \overline{V}_{h+1}^{k_i}(s_{h+1}^{k_i}) - V_{h+1}^\dagger(s_{h+1}^{k_i}) \right)$$
$$+ \sum_{i=1}^t \alpha_t^i \left( r_h^{k_i} - r_h(s, \widetilde{a}_h^{k_i}) + V_{h+1}^\dagger(s_{h+1}^{k_i}) - P_h V_{h+1}^\dagger(s, \widetilde{a}_h^{k_i}) + b_i \right) \tag{40}$$
$$+ \sum_{i=1}^t \alpha_t^i \left( Q_h^\dagger(s, \widetilde{a}_h^{k_i}) - Q_h^\dagger(s, \pi_h^-(s)) \right)$$

Since $\mathbb{E}[V_{h+1}^\dagger(s_{h+1}^k)|\mathcal{F}_h^k \cup \{s_h^k, a_h^k\}] = \mathbb{E}[V_{h+1}^\dagger(s_{h+1}^k)|s_h^k, a_h^k] = P_h V_{h+1}^\dagger(s, a)$ for any state-action pair $(s_h^k, a_h^k) = (s, a)$, $\sum_{i=1}^t \alpha_t^i \left( V_{h+1}^\dagger(s_{h+1}^{k_i}) - P_h V_{h+1}^\dagger(s, \widetilde{a}_h^{k_i}) \right)$ is the weighted sum of a martingale difference sequence w.r.t the filtration $\{\mathcal{F}_h^{k_i}\}_{i \geq 1}$. By Azuma-Hoeffding inequality, we have

$$\mathbb{P} \left( \left| \sum_{i=1}^t \alpha_t^i \left( V_{h+1}^\dagger(s_{h+1}^{k_i}) - P_h V_{h+1}^\dagger(s, \widetilde{a}_h^{k_i}) \right) \right| \geq \eta \right) \leq 2 \exp \left( -\frac{\eta^2}{2(H-h)^2 \frac{2H}{t}} \right), \tag{41}$$

where we used $\sum_{i=1}^t (\alpha_t^i)^2 \leq \frac{2H}{t}$ a property of $\alpha_t^i$. By setting the right hand side of the above equation to $p/(SAT)$ and $t = \overline{N}_h^k(s, a)$, we have for each fixed state-action pair $(s, a, h) \in \mathcal{S} \times \mathcal{A} \times [H]$, with probability at least $1 - p/(SAH)$, event $\mathcal{E}_1$ holds, where $\mathcal{E}_1$ is defined as

$$\mathcal{E}_1 := \{\forall k \in [K],$$
$$\left| \sum_{i=1}^t \alpha_t^i \left( V_{h+1}^\dagger(s_{h+1}^{k_i}) - P_h V_{h+1}^\dagger(s, \widetilde{a}_h^{k_i}) \right) \right| \leq (H-h) \sqrt{\frac{4H \log(2SAT/p)}{\overline{N}_h^k(s, a)}} \}. \tag{42}$$

Similarly, for each fixed state-action-step pair $(s, a, h) \in \mathcal{S} \times \mathcal{A} \times [H]$, with probability at least $1 - p/(SAH)$, we have event $\mathcal{E}_2$ holds with

$$\mathcal{E}_2 := \left\{ \forall k \in [K], \left| \sum_{i=1}^{t} \alpha_t^i \left( r_h^{k_i} - r_h(s, \tilde{a}_h^{k_i}) \right) \right| \leq \sqrt{\frac{4H \log(2SAT/p)}{\overline{N}_h^k(s, a)}} \right\}. \tag{43}$$

Then if the agent chooses $b_t = c\sqrt{H^3 \iota / t}$ for some constant $c$ and $\iota = \log(2SAT/p)$, by the property (3) of $\alpha_t^i$, we have $b_t \leq \sum_{i=1}^{t} \alpha_t^i b_i \leq 2b_t$. Under events $\mathcal{E}_1$ and $\mathcal{E}_2$, for $t \geq 1$, the third term of the RHS of equation (40) can be bounded by

$$(cH - 2(H - h) - 2)\sqrt{H\iota/t}$$
$$\leq \sum_{i=1}^{t} \alpha_t^i \left( r_h^{k_i} - r_h(s, \tilde{a}_h^{k_i}) + V_{h+1}^{\dagger}(s_{h+1}^{k_i}) - P_h V_{h+1}^{\dagger}(s, \tilde{a}_h^{k_i}) + b_i \right) \tag{44}$$
$$\leq (cH + 2(H - h) + 2)\sqrt{H\iota/t}.$$

For notational simplicity, let $\beta_h(t) = (cH + 2(H - h) + 2)\sqrt{H\iota/t}$ when $t > 0$ and $\beta_h(0) = 0$ for any step $h$.

We split the fourth term of the RHS of equation (40) into two cases.

If $\tilde{a}_h^k = \pi_h^{\dagger}(s)$ holds, we have

$$Q_h^{\dagger}(s, \tilde{a}_h^k) - Q_h^{\dagger}(s, \pi_h^-(s)) = Q_h^{\dagger}(s, \pi_h^{\dagger}(s)) - Q_h^{\dagger}(s, \pi_h^-(s)). \tag{45}$$

Let $B_h(t) = (e(H - h) + 1)\sqrt{\frac{2S\iota}{t}}$ when $t > 0$ and $B_h(0) = H$ for any step $h$.

If $\tilde{a}_h^k \neq \pi_h^{\dagger}(s)$ holds, the attacker attacked the agent, and from Lemma 2, we have with probability $1 - p$,

$$Q_h^{\dagger}(s, \pi_h^-(s)) \geq L_h^k(s, \pi_h^-(s)) \geq L_h^k(s, \tilde{a}_h^k) \geq Q_h^{\dagger}(s, \tilde{a}_h^k) - 2B_h\left(N_h^k(s, \tilde{a}_h^k)\right), \tag{46}$$

and $0 \leq Q_h^{\dagger}(s, \tilde{a}_h^k) - Q_h^{\dagger}(s, \pi_h^-(s)) \leq 2B_h\left(N_h^k(s, \tilde{a}_h^k)\right)$.

If $a = \pi_h^{\dagger}(s)$, the attacker does not attack so $\tilde{a}_h^{k_i} = a = \pi_h^{\dagger}(s)$. Then by combining (40) and (44), we have for $c \geq 2$

$$\sum_{i=1}^{t} \alpha_t^i \left( \overline{V}_{h+1}^{k_i}(s_{h+1}^{k_i}) - V_{h+1}^{\dagger}(s_{h+1}^{k_i}) \right) \leq \overline{Q}_h^k(s, \pi_h^{\dagger}(s)) - Q_h^{\dagger}(s, \pi_h^{\dagger}(s))$$
$$\leq \alpha_t^0 H + \sum_{i=1}^{t} \alpha_t^i \left( \overline{V}_{h+1}^{k_i}(s_{h+1}^{k_i}) - V_{h+1}^{\dagger}(s_{h+1}^{k_i}) \right) + \beta_h(t). \tag{47}$$

Since $\overline{V}_{H+1}^{k_i} = V_{H+1}^{\dagger} = 0$, from induction on $h = H, H - 1, \ldots, 1$, we have $\overline{V}_h^k(s) \geq \min\{\overline{Q}_h^k(s, \pi_h^{\dagger}(s)), H\} \geq V_h^{\dagger}(s)$ for all state $s$, step $h$ and episode $k$ with probability $1 - 2p$.

If $a \neq \pi_h^{\dagger}(s)$, the attacker attacks by changing the action to the target action or a possible worst action. From (40) and (44), we have

$$\overline{Q}_h^k(s, a) - Q_h^{\dagger}(s, \pi_h^-(s))$$
$$\leq \alpha_t^0 H + \sum_{i=1}^{t} \alpha_t^i \left( \overline{V}_{h+1}^{k_i}(s_{h+1}^{k_i}) - V_{h+1}^{\dagger}(s_{h+1}^{k_i}) \right) + \beta_h(t)$$
$$+ \sum_{i=1}^{t} \alpha_t^i \mathbb{1}\left( \tilde{a}_h^{k_i} = \pi_h^{\dagger}(s) \right) \left( Q_h^{\dagger}(s, \pi_h^{\dagger}(s)) - Q_h^{\dagger}(s, \pi_h^-(s)) \right) \tag{48}$$
$$+ \sum_{i=1}^{t} \alpha_t^i \mathbb{1}\left( \tilde{a}_h^{k_i} \neq \pi_h^{\dagger}(s) \right) 2B_h\left( N_h^{k_i}(s, \tilde{a}_h^{k_i}) \right),$$

and

$$\overline{Q}_h^k(s,a) - Q_h^\dagger(s, \pi_h^\dagger(s)) = \overline{Q}_h^k(s,a) - Q_h^\dagger(s, \pi_h^-(s)) - \Delta_h(s)$$

$$\leq \alpha_t^0 H + \sum_{i=1}^t \alpha_t^i \left( \overline{V}_{h+1}^{k_i}(s_{h+1}^{k_i}) - V_{h+1}^\dagger(s_{h+1}^{k_i}) \right) + \beta_h(t) \qquad (49)$$

$$+ \sum_{i=1}^t \alpha_t^i \mathbb{1} \left( \widetilde{a}_h^{k_i} \neq \pi_h^\dagger(s) \right) \left( 2B_h \left( N_h^{k_i}(s, \widetilde{a}_h^{k_i}) \right) - \Delta_h(s) \right),$$

where $\Delta_h(s) := Q_h^\dagger(s, \pi_h^\dagger(s)) - Q_h^\dagger(s, \pi_h^-(s))$.

## D.2    Proof of Theorem 3

In this section, we assume the two events $\mathcal{E}_1$, $\mathcal{E}_2$ hold. For any state $s \in \mathcal{S}$ and any step $h \in [H]$, Lemma 4 shows that in the agent's observations, $\overline{Q}_h^k(s, \pi_h^\dagger(s)) \geq Q_h^\dagger(s, \pi_h^\dagger(s))$ for all episodes $k \in [K]$ with probability $1 - 3p$. Since UCB-H takes action by the function $a_h^k = \arg\max_{a \in \mathcal{A}} \overline{Q}_h^k(s_h^k, a)$, we have that with probability $1 - 3p$, $\overline{Q}_h^k(s_h^k, a_h^k) \geq \overline{Q}_h^k(s_h^k, \pi_h^\dagger(s_h^k)) \geq Q_h^\dagger(s_h^k, \pi_h^\dagger(s_h^k))$ for all episodes $k \in [K]$ and all steps $h \in [H]$. Thus, we can bound the loss and cost functions by

$$\sum_{k=1}^K \sum_{h=1}^H \mathbb{1}\left( a_h^k \neq \pi^\dagger(s_h^k) \right) \Delta_h(s_h^k)$$

$$= \sum_{k=1}^K \sum_{h=1}^H \mathbb{1}\left( a_h^k \neq \pi^\dagger(s_h^k) \right) \left( Q_h^\dagger(s_h^k, \pi_h^\dagger(s_h^k)) - Q_h^\dagger(s_h^k, \pi_h^-(s_h^k)) \right) \qquad (50)$$

$$\leq \sum_{k=1}^K \sum_{h=1}^H \mathbb{1}\left( a_h^k \neq \pi^\dagger(s_h^k) \right) \left( \overline{Q}_h^k(s_h^k, a_h^k) - Q_h^\dagger(s_h^k, \pi_h^-(s_h^k)) \right).$$

First consider a fixed step $h$. The contribution of step $h$ to the loss function can be written as $\text{Loss}_h(K) = \sum_{k=1}^K \mathbb{1}\left( a_h^k \neq \pi^\dagger(s_h^k) \right)$. For notational convenience, denote

$$\phi_{h,h}^k := \mathbb{1}\left( a_h^k \neq \pi^\dagger(s_h^k) \right) \text{ and } \delta_h^k := \overline{Q}_h^k(s_h^k, a_h^k) - Q_h^\dagger \left( s_h^k, \pi_h^\dagger(s_h^k) \right). \qquad (51)$$

From the update equation of $V$-values in UCB-H algorithm, we have

$$\overline{V}_h^k(s_h^k) - \overline{V}_h^\dagger(s_h^k) = \min\{H, \max_{a \in \mathcal{A}} \overline{Q}_h^k(s_h^k, a)\} - \overline{V}_h^\dagger(s_h^k) \leq \delta_h^k. \qquad (52)$$

From Lemma 4, with probability $1 - 3p$, we have

$$\sum_{k=1}^K \phi_{h,h}^k \delta_h^k \leq \sum_{k=1}^K \phi_{h,h}^k \alpha_{\overline{N}_h^k(s_h^k, a_h^k)}^0 H + \sum_{k=1}^K \phi_{h,h}^k \beta_h \left( \overline{N}_h^k(s_h^k, a_h^k) \right)$$

$$+ \sum_{k=1}^K \phi_{h,h}^k \sum_{i=1}^{\overline{N}_h^k(s_h^k, a_h^k)} \alpha_{\overline{N}_h^k(s_h^k, a_h^k)}^i \delta_{h+1}^{k_i(s_h^k, a_h^k, h)}$$

$$+ \sum_{k=1}^K \phi_{h,h}^k \sum_{i=1}^{\overline{N}_h^k(s_h^k, a_h^k)} \alpha_{\overline{N}_h^k(s_h^k, a_h^k)}^i \mathbb{1}\left( \widetilde{a}_h^{k_i(s_h^k, a_h^k, h)} \neq \pi_h^\dagger(s_h^{k_i(s_h^k, a_h^k, h)}) \right) \cdot \qquad (53)$$

$$\left( 2B_h \left( N_h^{k_i(s_h^k, a_h^k, h)}(s_h^{k_i(s_h^k, a_h^k, h)}, \widetilde{a}_h^{k_i(s_h^k, a_h^k, h)}) \right) - \Delta_h(s_h^{k_i(s_h^k, a_h^k, h)}) \right),$$

where $k_i(s, a, h)$ represents the episode where $(s, a)$ was taken by the agent at step $h$ for the $i$th time.

The key step is to upper bound the third term in the RHS of (53). Note that for any episode $k$, the third term takes all the prior episodes $k_i < k$ where $(s_h^k, a_h^k)$ was taken into account. In other words, for any episode $k'$, the term $\delta_{h+1}^{k'}$ appears in the second term at all posterior episodes $k > k'$ where $(s_h^{k'}, a_h^{k'})$

was taken. The first time it appears we have $\overline{N}_h^k(s_h^k, a_h^k) = \overline{N}_h^k(s_h^{k'}, a_h^{k'}) = \overline{N}_h^{k'}(s_h^{k'}, a_h^{k'}) + 1$ and the second time it appears we have $\overline{N}_h^k(s_h^k, a_h^k) = \overline{N}_h^k(s_h^{k'}, a_h^{k'}) = \overline{N}_h^{k'}(s_h^{k'}, a_h^{k'}) + 2$, and so on. Thus, we exchange the order of summation and have

$$
\sum_{k=1}^{K} \phi_{h,h}^k \sum_{i=1}^{\overline{N}_h^k(s_h^k, a_h^k)} \alpha_{\overline{N}_h^k(s_h^k, a_h^k)}^i \delta_{h+1}^{k_i(s_h^k, a_h^k, h)}
$$
$$
= \sum_{k'=1}^{K} \delta_{h+1}^{k'} \sum_{t=\overline{N}_h^{k'}(s_h^{k'}, a_h^{k'})+1}^{\overline{N}_h^K(s_h^{k'}, a_h^{k'})} \phi_{h,h}^{k_t(s_h^{k'}, a_h^{k'}, h)} \alpha_t^{\overline{N}_h^{k'}(s_h^{k'}, a_h^{k'})+1}. \tag{54}
$$

For any $k \in [K]$, let $\phi_{h,h+1}^k = \sum_{t=\overline{N}_h^k(s_h^k, a_h^k)+1}^{\overline{N}_h^K(s_h^k, a_h^k)} \phi_{h,h}^{k_t(s_h^k, a_h^k)} \alpha_t^{\overline{N}_h^k(s_h^k, a_h^k)+1}$. The third term in the RHS of (53) can be simplified as $\sum_{k=1}^{K} \phi_{h,h+1}^k \delta_{h+1}^k$. The fourth term in the RHS of (53) can be simplified as

$$
\sum_{k=1}^{K} \phi_{h,h+1}^k \mathbb{1}\left(\widetilde{a}_h^k \neq \pi_h^\dagger(s_h^k)\right)\left(2B_h\left(N_h^k(s_h^k, \widetilde{a}_h^k)\right) - \Delta_h(s_h^k)\right). \tag{55}
$$

Since $\alpha_t^0 = 0$ when $t \geq 1$, $\sum_{k=1}^{K} \phi_{h,h}^k \alpha_{\overline{N}_h^k(s_h^k, a_h^k)}^0 H \leq SAH$. Thus, we can rewrite (53) as

$$
\sum_{k=1}^{K} \phi_{h,h}^k \delta_h^k \leq SAH + \sum_{k=1}^{K} \phi_{h,h+1}^k \delta_{h+1}^k + \sum_{k=1}^{K} \phi_{h,h}^k \beta_h\left(\overline{N}_h^k(s_h^k, a_h^k)\right)
$$
$$
+ \sum_{k=1}^{K} \phi_{h,h+1}^k \mathbb{1}\left(\widetilde{a}_h^k \neq \pi_h^\dagger(s_h^k)\right)\left(2B_h\left(N_h^k(s_h^k, \widetilde{a}_h^k)\right) - \Delta_h(s_h^k)\right). \tag{56}
$$

Recursing the result for $h' = h, h+1, \ldots, H$, and using the fact $\delta_{H+1}^k = 0$ for all episode $k$, we have

$$
\sum_{k=1}^{K} \phi_{h,h}^k \delta_h^k \leq SAH(H - h + 1) + \sum_{h'=h}^{H} \sum_{k=1}^{K} \phi_{h,h'}^k \beta_{h'}\left(\overline{N}_{h'}^k(s_{h'}^k, a_{h'}^k)\right)
$$
$$
+ \sum_{h'=h}^{H} \sum_{k=1}^{K} \phi_{h,h'+1}^k \mathbb{1}\left(\widetilde{a}_{h'}^k \neq \pi_{h'}^\dagger(s_h^k)\right) 2B_{h'}\left(N_{h'}^k(s_{h'}^k, \widetilde{a}_{h'}^k)\right) \tag{57}
$$
$$
- \sum_{h'=h}^{H} \sum_{k=1}^{K} \phi_{h,h'+1}^k \mathbb{1}\left(\widetilde{a}_{h'}^k \neq \pi_{h'}^\dagger(s_h^k)\right) \Delta_h(s_{h'}^k).
$$

Here, we present some important properties of $\phi_{h,h'}^k$ for all step $h' \geq h$ when step $h$ are fixed:

(1) $\sum_{k=1}^{K} \phi_{h,h}^k = \sum_{k=1}^{K} \mathbb{1}\left(a_h^k \neq \pi^\dagger(s)\right) = \text{Loss}_h(K)$;

(2) $\sum_{k=1}^{K} \phi_{h,h'}^k = \sum_{k=1}^{K} \phi_{h,h}^k$, for all step $h' \geq h$;

(3) $\max_{k \in [K]} \phi_{h,h'+1}^k \leq (1 + \frac{1}{H}) \max_{k \in [K]} \phi_{h,h'}^k$ for all step $h' \geq h$;

(4) $\max_{k \in [K]} \phi_{h,h}^k = 1$, and $\max_{k \in [K]} \phi_{h,h'}^k \leq e$ for all step $h' \geq h$.

Property (1) is from the definition of $\overline{N}_h^k(s)$. Properties (2) and (3) can be proved by the properties of $\alpha_t^i$. In particular, for all step $h' \geq h$,

$$
\sum_{k=1}^{K} \phi_{h,h'+1}^k = \sum_{k=1}^{K} \phi_{h,h'}^k \sum_{i=1}^{\overline{N}_{h'}^k(s_{h'}^k, a_{h'}^k)} \alpha_{\overline{N}_{h'}^k(s_{h'}^k, a_{h'}^k)}^i = \sum_{k=1}^{K} \phi_{h,h'}^k, \tag{58}
$$

and for all step $h' \geq h$ and all episode $k \in [K]$,

$$
\phi_{h,h'+1}^k = \sum_{t=\overline{N}_{h'}^k(s_{h'}^k,a_{h'}^k)+1}^{\overline{N}_{h'}^K(s_{h'}^k,a_{h'}^k)} \phi_{h,h'}^{k_t(s_{h'}^k,a_{h'}^k,h')} \alpha_t^{\overline{N}_{h'}^k(s_{h'}^k,a_{h'}^k)+1}
$$

$$
\leq \sum_{t=\overline{N}_{h'}^k(s_{h'}^k,a_{h'}^k)+1}^{\overline{N}_{h'}^K(s_{h'}^k,a_{h'}^k)} \alpha_t^{\overline{N}_{h'}^k(s_{h'}^k,a_{h'}^k)+1} \max_{k\in[K]} \phi_{h,h'}^k \tag{59}
$$

$$
\leq (1 + \frac{1}{H}) \max_{k\in[K]} \phi_{h,h'}^k.
$$

Property (4) is from Property (3) and the fact $(1 + \frac{1}{H})^H \leq e$.

Now we are ready to prove Theorem 3. At first, we bound the second term of the RHS of (57). We regroup the summands in a different way.

$$
\sum_{h'=h}^H \sum_{k=1}^K \phi_{h,h'}^k \cdot \beta_{h'} \left( \overline{N}_{h'}^k(s_{h'}^k,a_{h'}^k) \right) = \sum_{h'=h}^H \sum_{(s,a)\in\mathcal{S}\times\mathcal{A}} \sum_{t=1}^{\overline{N}_{h'}^K(s,a)} \phi_{h,h'}^{k_t(s,a,h')} \beta_{h'}(t-1)
$$

$$
= \sum_{h'=h}^H \sum_{(s,a)\in\mathcal{S}\times\mathcal{A}} \sum_{t=2}^{\overline{N}_{h'}^K(s,a)} \phi_{h,h'}^{k_t(s,a,h')} \beta_{h'}(t-1), \tag{60}
$$

because $\beta_{h'}(0) = 0$. Define $\phi_{h,h'}^{(s,a)} = \sum_{t=1}^{\overline{N}_{h'}^K(s,a)} \phi_{h,h'}^{k_t(s,a,h)}$. Since $\sqrt{\frac{1}{t}}$ is a monotonically decreasing positive function for $n \geq 1$ and $\phi_{h,h'}^{k_t(s,a,h')} \leq e$, by the rearrangement inequality, for $h' \geq h$, we have

$$
\sum_{t=1}^{\overline{N}_{h'}^K(s,a)} \phi_{h,h'}^{k_t(s,a,h)} \sqrt{\frac{1}{t}} \leq \sum_{t=1}^{\lfloor\phi_{h,h'}^{(s,a)}/e\rfloor} e\sqrt{\frac{1}{t}} + (\phi_{h,h'}^{(s,a)} - \lfloor\phi_{h,h'}^{(s,a)}/e\rfloor)\sqrt{\frac{1}{\lceil\phi_{h,h'}^{(s,a)}/e\rceil}}
$$

$$
\leq e\sqrt{\frac{1}{1}} + \int_1^{\phi_{h,h'}^{(s,a)}/e} e\sqrt{\frac{1}{t}}dt \leq 2\sqrt{e\phi_{h,h'}^{(s,a)}}. \tag{61}
$$

By plugging (61) back into (60) we have

$$
\sum_{h'=h}^H \sum_{k=1}^K \phi_{h,h'}^k \cdot \beta_{h'} \left( \overline{N}_{h'}^k(s_{h'}^k,a_{h'}^k) \right) \leq \sum_{h'=h}^H 2(cH + 2(H-h') + 2)\sqrt{eSAH\iota \sum_{k=1}^K \phi_{h,h}^k}
$$

$$
\leq H(cH + 2H + 2)\sqrt{eSAH\iota \sum_{k=1}^K \phi_{h,h}^k} \tag{62}
$$

$$
= H(cH + 2H + 2)\sqrt{eSAH\iota\mathrm{Loss}_h(K)},
$$

where the first inequality holds due to $\sum_{(s,a)\in\mathcal{S}\times\mathcal{A}} \phi_{h,h'}^{(s,a)} = \sum_{k=1}^K \phi_{h,h'}^k$ and $\sqrt{t}$ is a concave function for $t \geq 0$.

Similarly, we can bound a part of the third term of the RHS of (57) by

$$\sum_{h'=h}^{H} \sum_{k=1}^{K} \phi_{h,h'+1}^{k} \mathbb{1}\left(\widetilde{a}_{h'}^{k} \neq \pi_{h'}^{\dagger}(s_{h}^{k})\right) 2B_{h'}\left(N_{h'}^{k}(s_{h'}^{k}, \widetilde{a}_{h'}^{k})\right)$$

$$\overset{①}{\leq} \sum_{h'=h}^{H} \sum_{(s,\widetilde{a})\in\mathcal{S}\times\mathcal{A}} \sum_{t=1}^{N_{h'}^{K}(s,\widetilde{a})} \phi_{h,h'+1}^{k_t(s,\widetilde{a},h')} 2B_{h'}(t-1)$$

$$\overset{②}{\leq} \sum_{h'=h}^{H} \sum_{(s,\widetilde{a})\in\mathcal{S}\times\mathcal{A}} \sum_{t=2}^{N_{h'}^{K}(s,\widetilde{a})} \phi_{h,h'+1}^{k_t(s,\widetilde{a},h')} 2B_{h'}(t-1) + 2e(H-h+1)SAH \tag{63}$$

$$\overset{③}{\leq} H^2 e \sqrt{SA\iota \sum_{k=1}^{K} \phi_{h,h}^{k}} + 2e(H-h+1)SAH$$

$$= H^2 e \sqrt{SA\iota \text{Loss}_h(K)} + 2e(H-h+1)SAH$$

where $k_t(s,\widetilde{a},h)$ represents the episode where $(s,\widetilde{a})$ was taken by the attacker at step $h$ for the $i$th time. Here, ① comes from deleting the indicator function and regrouping the summands; ② follows $\phi_{h,h'}^{k} \leq e$ and $B_h(0) = H$; ③ follows the same steps in (61) and (62).

As shown in (50), we have

$$0 \leq \sum_{k=1}^{K} \phi_{h,h}^{k} \left(\overline{Q}_h^k(s_h^k, a_h^k) - Q_h^{\dagger}\left(s_h^k, \pi_h^{-}(s_h^k)\right)\right) - \sum_{k=1}^{K} \phi_{h,h}^{k} \Delta_h(s_h^k) \leq \sum_{k=1}^{K} \phi_{h,h}^{k} \delta_h^k. \tag{64}$$

Thus, we need to find the lower bound of the fourth term of the RHS of (57). Since $\Delta_h(s_h^k) > \Delta_{min} > 0$, we have

$$\sum_{h'=h}^{H} \sum_{k=1}^{K} \phi_{h,h'+1}^{k} \mathbb{1}\left(\widetilde{a}_{h'}^{k} \neq \pi_{h'}^{\dagger}(s_h^k)\right) \Delta_h(s_{h'}^k)$$

$$\geq \sum_{k=1}^{K} \phi_{h,h+1}^{k} \mathbb{1}\left(\widetilde{a}_h^k \neq \pi_h^{\dagger}(s_h^k)\right) \Delta_h(s_h^k)$$

$$\geq \Delta_{min} \sum_{k=1}^{K} \phi_{h,h+1}^{k} \mathbb{1}\left(\widetilde{a}_h^k \neq \pi_h^{\dagger}(s_h^k)\right) \tag{65}$$

$$= \Delta_{min} \left(\text{Loss}_h(K) - \sum_{k=1}^{K} \phi_{h,h+1}^{k} \mathbb{1}\left(\widetilde{a}_h^k = \pi_h^{\dagger}(s_h^k)\right)\right).$$

Recall the definition of $\phi_{h,h+1}^{k}$ and the property (5) of $\alpha_t^i$, we have

$$\sum_{k=1}^{K} \phi_{h,h+1}^{k} \mathbb{1}\left(\widetilde{a}_h^k = \pi_h^{\dagger}(s_h^k)\right)$$

$$= \sum_{k=1}^{K} \sum_{t=\overline{N}_h^k(s_h^k, a_h^k)+1}^{\overline{N}_h^K(s_h^k, a_h^k)} \phi_{h,h}^{k_t(s_h^k, a_h^k, h)} \alpha_t^{\overline{N}_h^k(s_h^k, a_h^k)+1} \mathbb{1}\left(\widetilde{a}_h^k = \pi_h^{\dagger}(s_h^k)\right)$$

$$= \sum_{s\in\mathcal{S}} \sum_{k=1}^{K} \mathbb{1}\left(s_h^k = s\right) \mathbb{1}\left(\widetilde{a}_h^k = \pi_h^{\dagger}(s)\right) \mathbb{1}\left(a_h^k \neq \pi_h^{\dagger}(s)\right) \sum_{t=\overline{N}_h^k(s, a_h^k)+1}^{\overline{N}_h^K(s, a_h^k)} \alpha_t^{\overline{N}_h^k(s, a_h^k)+1} \tag{66}$$

$$\leq (1+\frac{1}{H}) \sum_{k=1}^{K} \mathbb{1}\left(\widetilde{a}_h^k = \pi_h^{\dagger}(s_h^k)\right) \mathbb{1}\left(a_h^k \neq \pi_h^{\dagger}(s_h^k)\right).$$

Recall the inequality (30). We have with probability $1 - p$, for all $h \in [H]$

$$
\sum_{k=1}^{K} \mathbb{1}\left(a_h^k \neq \pi_h^\dagger(s_h^k)\right) \mathbb{1}\left(\widetilde{a}_h^k \neq \pi_h^\dagger(s)\right)
$$
$$
\geq \frac{1}{H} \sum_{k=1}^{K} \mathbb{1}\left(a_h^k \neq \pi_h^\dagger(s_h^k)\right) - \sqrt{2log(2H/p) \sum_{k=1}^{K} \mathbb{1}\left(a_h^k \neq \pi_h^\dagger(s_h^k)\right)},
$$
(67)

which is equivalent to

$$
\sum_{k=1}^{K} \mathbb{1}\left(a_h^k \neq \pi_h^\dagger(s_h^k)\right) \mathbb{1}\left(\widetilde{a}_h^k = \pi_h^\dagger(s)\right)
$$
$$
\leq (1 - \frac{1}{H}) \sum_{k=1}^{K} \mathbb{1}\left(a_h^k \neq \pi_h^\dagger(s_h^k)\right) + \sqrt{2log(2H/p) \sum_{k=1}^{K} \mathbb{1}\left(a_h^k \neq \pi_h^\dagger(s_h^k)\right)},
$$
(68)

Plugging these back into (66) and further (65), we have

$$
\sum_{h'=h}^{H} \sum_{k=1}^{K} \phi_{h,h'+1}^k \mathbb{1}\left(\widetilde{a}_{h'}^k \neq \pi_{h'}^\dagger(s_h^k)\right) \Delta_h(s_{h'}^k)
$$
$$
\geq \Delta_{min}\left(\frac{1}{H^2}\text{Loss}_h(K) - (1 + \frac{1}{H})\sqrt{2log(2H/p) \sum_{k=1}^{K} \text{Loss}_h(K)}\right).
$$
(69)

Combining (57), (62), (63) and (69), we have

$$
\Delta_{min}\left(\frac{1}{H^2}\text{Loss}_h(K) - (1 + \frac{1}{H})\sqrt{2log(2H/p) \sum_{k=1}^{K} \text{Loss}_h(K)}\right)
$$
$$
\leq H^2 e \sqrt{SA\iota \text{Loss}_h(K)} + 2e(H - h + 1)SAH
$$
$$
+ SAH(H - h + 1) + H(cH + 2H + 2)\sqrt{eSAH\iota \text{Loss}_h(K)},
$$
(70)

which is equivalent to

$$
\text{Loss}_h(K) \leq 2(H^2 + H)^2 \log(2H/p) + \frac{1}{\Delta_{min}} SAH^2(H - h + 1)
$$
$$
+ \frac{1}{\Delta_{min}^2} e^2 H^8 SA\iota + \frac{1}{\Delta_{min}^2} eH^7(cH + 2H + 2)^2 SA\iota.
$$
(71)

This establishes

$$
\text{Cost}(K, H) \leq \text{Loss}(K) \leq O(H^5 \log(2H/p) + \frac{1}{\Delta_{min}} SAH^4 + \frac{1}{\Delta_{min}^2} H^{10} SA\iota). \quad (72)
$$