# OpenReview forum: "Provably Efficient Black-Box Action Poisoning Attacks Against Reinforcement Learning"
_NeurIPS.cc/2021/Conference — NeurIPS 2021 Poster_

### Official Review · Reviewer_R4FA · 2021-07-08

**Rating:** 6
**Confidence:** 3

**Summary:**

The paper studies poisoning attacks in reinforcement learning that override an agent's actions in order to force the policy of interest (target policy). The problem of 'action poisoning' is formulated in an episodic reinforcement learning framework, where the attacker experiences the cost of poisoning actions (equal to the number of poisoned/changed actions) and the loss of not fully forcing the target policy (equal to the number of times target policy was not followed). The paper formally studies white-box and black-box poisoning, developing a principled way of performing action poisoning attacks with provable guarantees. The main algorithmic contribution of this paper is an algorithm called LCB-H that has provable bounds on the cost and loss of poisoning. The numerical experiments are conducted to demonstrate the efficacy of the proposed black box attack.

**Limitations And Societal Impact:**

The limitations are stated in Section 5. Potential societal impacts are discussed in the introduction.

**Main Review:**

a. Originality

To my knowledge, this is the first paper to formally study action poisoning attacks in episodic tabular RL. That being said, [Sun et al. 2021] discuss/study different 'poisoning aims' including poisoning actions (see Section 3.3.1). In that sense, some of the claims in the paper do not seem to be precise, e.g., 'Prior works on adversarial attacks in RL mainly focus on either observation poisoning attacks or environment poisoning attacks'; the same is true for the claimed contributions. Other related work is generally covered well, but could be expanded with a few more references. For example, while the introduction mentions more than 10 references on adversarial attacks in supervised learning, test-time attacks in RL are not covered to a great extent. There are also references on poisoning attacks in RL that could be added, including some older works on policy teaching, e.g., Zhang and Parkes 'Value-Based Policy Teaching with Active Indirect Elicitation'.

b. Quality

The paper studies an important topic and highly relevant for designing robust RL systems. The results are interesting as they provide a principled way of designing action poisoning attacks in tabular RL setting. On the other hand, it is not clear how practical the proposed approach is. I also have some concerns/questions about the modeling assumptions.

b.1.  The paper claims that this form of poisoning attacks is more practical than other forms of poisoning attacks, and the arguments are based on 'restrictiveness' ('Compared with existing attack models, the attacker’s ability in the proposed action poisoning  attack model is more restricted, and hence the attack model is more practical. '). I don't see why 'restrictiveness' would necessarily imply that the attack is more practical. Whereas action poisoning could be prevented by increasing the security of the agent, poisoning rewards might not be since rewards can be external to the agent (e.g., feedback in the recommender systems). In this sense, action poisoning might not be practical at all in some settings since it assumes a (partial) control over the agent.

b.2. The overall goal could be more clearly specified and ideally formulated as an optimization problem. In Section 2, we have 'The attacker aims to minimize both the attack cost and the loss of attacks, or minimize one of them subject to a constraint on the one another.'. However, it seems that the true goal is to have provable upper bounds on the cost/loss of attacks. The paper could also motivate the cost/loss function. E.g., in the car driving scenario from the introduction, why would it be more costly to manipulate the signal N times than M times? Why wouldn't we model the loss of the attacker through its own reward function?

b.3. The proposed algorithm looks interesting and has provable guarantees on its performance.  It is less clear whether the bounds are tight and/or informative for practical purposes. For example, some of the bounds have dependency on $H^10$ (Theorem 3), which means that even for smaller values of $H$, the bounds are quite large. The arguments regarding UCB-H and Theorem 3 are somewhat unclear. Theorem 2 considers dynamic regret, whereas the arguments are based on the results from [Yang et al., 2021], which seems to consider the notion of regret similar to (4).

b.4. Experiments are relative simple, which is not surprising given that this is a theory paper. However, the running times indicate that the proposed approach might not be scalable. It is generally hard to follow the details in Section 4: Figure 1 has typos (Tiem -> Time?), y-axis is not labelled (and represent multiple measures), and the font is too small, the text mentions three agents, but it doesn't explain which ones (presumably, these are the agents in the subtitles of Figure 1?).

c. Clarity

The paper is enjoyable to read. However, there are many typos and some parts could be improved and explained in more detail. A few examples:
- The statement of Lemma 1 is not clear: what does it mean 'in the observation of the agent'?
- In Section 4, the statements 'Ten states can be considered as a circle' and 'sampled randomly from the five adjacent states' are not clear.
- The organization could be improved as well. For example, Assumption 1 and Eq. 4 could be a part of the setting (otherwise, it's not clear whether this assumption applies to Section 3.1 only). In Section 2, the claim is that the performance of the agent is measured with regret as defined by (2), but then in Section 3.2 we have a different notion of the regret.
- What is $\pi_{h}^{-}$ in line 219?
- The notion of behavior policy $b_k$ could be explained before being used in 246.

d. Significance

In my opinion, these results will be interesting to researchers working on adversarial attack in RL, and more specifically on poisoning attacks. The work is mainly theoretical, but some of the principles presented in the paper might lead to practical solutions. Compared to prior work, the paper studies a different setting and orthogonal aspects of poisoning attacks. Overall, I believe that the paper provides a good starting point for formally analyzing action poisoning attacks.

**Time Spent Reviewing:**

4h/5h

---

> ### Author Response · Authors · 2021-08-10
> **Reply to the reviewer's question**
>
> We thank Reviewer R4FA for the constructive and detailed feedback. We address some questions raised by Reviewer R4FA as follows:
>
> a. Originality
>
> Reply: We thank Reviewer R4FA for the positive feedback on the originality and novelty of our paper. We agree with the reviewer that some of the claims in the paper may not be precise. We will fix them in the revised version. We will also add more references about test-time attacks and policy teaching in our revised paper.
>
> b. Quality
>
> Reply to b.1: We will fix the argument about the relationship between 'restrictive' and 'practical', as they are not necessarily connected. We believe that the action poisoning attacks are practical. In some applications of RL models, action decisions and reward signals may need to be transmitted over communication links. For example, in autonomous vehicles, decisions are made at central control units (CPU) and these decisions will need to be transmitted to various parts of the vehicle through communication links. When data packets containing the reward signals and action decisions etc are transmitted through the communication links, an attacker can implement adversarial attacks by intercepting and modifying these data packets. The attacker does not need to control the agent, it just needs to intercept the communication links or control the buffer where the action signal is stored. Hence, our action poisoning attack is possible.
>
> We note that the goal of this paper is not to promote action manipulation attacks. Our goal is to understand the potential risks of action manipulation attacks. Understanding the risks of different kinds of adversarial attacks on RL is essential for the safe applications of RL model and designing robust RL systems. Our study reveals that, with small costs, action manipulation attacks can indeed lead the agent to suboptimal policies. We hope that our study could lead to more follow-up work (either by ourselves or other researchers) to make RL algorithms robust to such types of attacks.
>
> Reply to b.2: We agree with the reviewer that the overall goal could be formulated as an optimization problem such as
> $$
> min_{\vec{\widetilde{a}} \in \mathcal{A}^{K \times H}}~Cost(K,H) + \lambda Loss(K,H),
> $$
> or
>
> $$
> min_{\vec{\widetilde{a}} \in \mathcal{A}^{K \times H}} ~ Loss(K,H)
> $$
> $$
>  s.t.~  Cost(K,H) < C,
> $$
> where $\vec{\widetilde{a}}$ is the sequence of all post-attack actions ($\widetilde{a}_h^k$ for all $h$ from $1$ to $H$ and all $k$ from $1$ to $K$), $\lambda$ is a weight parameter and C is the attack budget.
>
> However, obtaining optimal solutions to these optimization problems is challenging. As the first step towards understanding the impact of action poisoning attacks, we design some specific simple yet effective attack strategies. In other words, in this paper, we aim to design white-box black-box attack strategies whose cost of attacks can be upper bounded. It will be interesting to further derive a lower bound of the sum of attack cost and loss so that we can know how far away our schemes are from optimal solutions.
>
> The main reason that we use the number of action modifications as the cost is that it is a natural way to measure the attacker's efforts. In addition, the less frequently the attacker manipulates the action signals, the more difficult for the attack to be detected. Hence, we define the cost function by the number of manipulation. We have also considered designing the loss of the attacker function through the reward function or the state transition probability. However, we think that the similarity of the reward function does not necessarily reflect the similarity of the action. In the other words, although the reward of an action is close to the reward of the target action, it may not be the attacker's target. Thus, as shown in the definition of the loss function in our paper, we consider the loss of any non-target action as $1$.
>
> Reply to b.3:  As shown in Theorem 1, the upper bound of the expected cost and loss of the $1/H$-portion attack in the white-box attack case is linear dependent on $H$. In this white-box attack case, the algorithm is practical even though the horizon is long. In the more practical black-box attack case, our LCB-H attack scheme nearly matches the performance of the $1/H$-portion white-box attacks, except with an additional term $307SAH^4\log(2SAT/p)/\Delta_{min}$. This additional term is dependent on how we derive the lower confidence bound. In our current paper, this bound is derived using Hoeffding-type martingale concentration inequalities. This bound can be improved by using Bernstein-type concentration inequalities. Furthermore, in our experiment results, we observe that the cost and loss of $1/H$-portion white-box attacks are about $H/\Delta_{min}$ times as much as the regret, and the performance of LCB-H black-box attacks is very close to that of the $1/H$-portion attacks, which suggests that the true performance of the designed black-box attack is very similar to the white-box attack strategy whose dependence of $H$ is only linear. Thus, we believe that our algorithm is practical when the horizon is not too long.
>
> In Theorem 2, we consider dynamic regret. However, the UCB-H is a stationary RL algorithm that does not have no-regret guarantees in a non-stationary setting. To show the effects of the proposed LCB-H action poisoning attack strategy, we analyzed the cost and loss of LCB-H action attacks on UCB-H. The results show that LCB-H attack is able to force UCB-H agent to choose actions by spending only logarithm cost. Due to the regret bound of UCB-H have a dependency on $H^6$ from [Yang et al., 2021], the bound in Theorem 3 has a high dependency on $H$. If an algorithm whose dynamic regret bound scales as $O(H^6SA log(T)/\Delta)$, the cost of LCB-H attacks on it should scales as $O(H^7SA log(T)/\Delta)$. Our results show that the cost of LCB-H attacks on UCB-H scales as $O(H^{10}SA log(T)/\Delta)$. We believe that UCB-H has a dynamic regret bound that scales as $O(H^9SA log(T)/\Delta)$ in this action poisoning environment. We will make the claim more clear in the revised version.
>
> Reply to b.4: The running time of the experiment consists of three parts: the agent, the environment, and the attacker. Most of the running time is spent on the random number generation, which is part of the environment. The LCB-H attack scheme is time efficient and is only linear dependent on time step $t$. As discussed in the reply to your comment b.3, our algorithm is practical even though the horizon is long.
> As we described at the beginning of Section 4, the three agents are UCB-H [Jin et al., 2018], UCB-B [Jin et al., 2018] and UCBVI-CH [Azar et al., 2017].
> The y-axis of the figure represents the cumulative cost/loss/regret from beginning to time $t$.
> We will fix the typos and make the figure more clear in the revised version.
>
> c. Clarity
>
> Comment: The statement of Lemma 1 is not clear: what does it mean 'in the observation of the agent'?
>
> Reply: As mentioned in Section 3.1, the combination of the attacker and the environment can be considered as a new environment to the agent. Because the agent does not know the presence of the attacker and the $\alpha$-portion action scheme is fixed over the time steps, the observation of the agent is a new stationary environment $M'$. We will clarify this in the revised paper.
>
> Comment: In Section 4, the statements 'Ten states can be considered as a circle' and 'sampled randomly from the five adjacent states' are not clear.
>
> Reply: We consider a $1 \times 10$ grid world. When the agent is at the leftmost state (s=1), it is possible for the next state to transit to the rightmost state (s=10). Similarly, when the agent is at the rightmost state (s=10), it is possible for the state to transit to the leftmost state (s=1). In this sense, these ten states can be considered as a circle, as these ten states are connected as a loop.
>
> By 'sampled randomly from the five adjacent states', we mean that the next state is a random mapping of the agent's action. We use this to simulate the stochastic environment $P(s'|s,a)$. For example, when, at state $s=5$, the environment receives the action $a =$ 'one step left',  the transition probabilities $P(3|5,one\~step\~ left)=P(5|5,one\~step\~left)=P(6|5,one\~step\~left)=P(7|5,one\~step\~left)=(1-p)/5$, $P(4|5,one\~step\~left)=p+(1-p)/5$ and other probabilities are equal to 0.
>
> Comment: The organization could be improved as well. For example, Assumption 1 and Eq. 4 could be a part of the setting (otherwise, it's not clear whether this assumption applies to Section 3.1 only). In Section 2, the claim is that the performance of the agent is measured with regret as defined by (2), but then in Section 3.2 we have a different notion of the regret.
>
> Reply: We will fix the organization and make it more readable in the revised version.
>
> Comment: What is $\pi_h^-$ in line 219?
>
> Reply: For a given target policy $\pi^+$, we define $\pi_h^-=argmin_{a}~Q_h^{+}(s,a)$. The definition is described in line 201. We will fix the organization and make this more clear in the revised paper.
>
> Comment: The notion of behavior policy $b_k$ could be explained before being used in 246.
>
> Reply: Thank you for the notification. We will fix it.
>
> d. Significance
>
> Comment: Thank you for the positive feedback. We hope our work may provide some ideas to build safe RL applications and design robust RL systems.
>
> Refs:
>
> [1] Yanchao Sun, Da Huo, and Furong Huang. Vulnerability-aware poisoning mechanism for online RL with unknown dynamics. In International Conference on Learning Representations, 2021.
>
> [2] Chi Jin, Zeyuan Allen-Zhu, Sebastien Bubeck, and Michael I Jordan. Is q-learning provably efficient? In Advances in Neural Information Processing Systems, volume 31, 2018.
>
> [3] Mohammad Gheshlaghi Azar, Ian Osband, and Rémi Munos. Minimax regret bounds for reinforcement learning. In International Conference on Machine Learning, pages 263–272, 2017.

---

### Official Review · Reviewer_L6CC · 2021-07-16

**Rating:** 6
**Confidence:** 3

**Summary:**

This paper proposes a new action poisoning framework to mislead the agent to learn a target policy. In a white-box setting, the paper introduces $\alpha$-portion attack that is guaranteed to conduct a successful attack with sublinear cost. In a black-box setting, the paper proposes an adaptive attack scheme called LCB-H that nearly matches the performance guarantees of the white-box attack method. With a case study on attacking UCB-H, the authors show that the proposed attack strategy can manipulate the learner with a logarithm cost, matching the main claim of the paper.

**Limitations And Societal Impact:**

The authors have explicitly discussed the limitations of the work in Section 5, as well as some future directions in Section 6. I would be curious to see how the proposed algorithm applies to state or reward poisoning, and a large-scale environment with function approximation.

**Main Review:**

The authors formally define and discuss action poisoning in multiple settings. The idea of attacking with the lower confidence bound is novel. The proposed action poisoning attack can deal with a practical black-box scenarion with log(T) cost, which is an interesting and significant contribution. The paper is in general well-written and easy to follow. The assumptions and limitations of the work are clearly stated, so it is easy to know the scope of the work.

My main concerns are: (1) how realistic are action poisoning attacks? Although the paper motivates the action attacks with auto-driving systems, the algorithm and analysis are built on a tabular case, where I find action attacks not very practical and easy to be detected.
(2) the cost of the poisoning algorithm, although is logarithm in T, has a high dependence on H, and would be large as the target policy is closer to the worst policy. Does it suggest that the algorithm becomes impractical when the horizon is long and when the target policy is close, though not equal, to the worst policy?

**Time Spent Reviewing:**

5

---

> ### Author Response · Authors · 2021-08-10
> **Reply to the reviewer's question**
>
> We thank Reviewer L6CC for the positive and valuable feedback. We address some questions raised by Reviewer L6CC as follows:
>
> 1. Comment: How realistic are action poisoning attacks? Although the paper motivates the action attacks with auto-driving systems, the algorithm and analysis are built on a tabular case, where I find action attacks not very practical and easy to be detected.
>
> Reply: Yes, as the first step of understanding the impact of action poisoning attacks, our analysis is focused on the tabular case. We are currently working on extending the results to the case with continuous state space. Similar to other studies on RL, extending the work from the tabular case to the continuous case is not straightforward. At the same time, even in the tabular case, it is not easy to detect the attack, because the policy selected by the attacker is the optimal policy from the agent's perspective. Furthermore, in the tabular model, the reward and next state can be generated randomly. In other words, the reward functions and the state transferring are not deterministic. Hence, if the agent does not know the true underlying MDP, which is the case in most RL applications, it is not straightforward to detect such attacks. In any case, the goal of our paper is not to promote action poisoning attacks. Rather, our goal is to understand and identify the impacts of such attacks. We hope our work can inspire follow-up work that can detect and mitigate such attacks so that RL models can be used in safety-critical applications.
>
> 2. Comment: The cost of the poisoning algorithm, although is logarithm in $T$, has a high dependence on $H$, and would be large as the target policy is closer to the worst policy. Does it suggest that the algorithm becomes impractical when the horizon is long and when the target policy is close, though not equal, to the worst policy?
>
> Reply: As shown in Theorem 1, the upper bound of the expected cost and loss of the $1/H$-portion attack in the white-box attack case is linear dependent on $H$. Hence, in the white-box attack case, the algorithm is practical even though the horizon is long.
>
> In the more practical black-box attack case, our LCB-H attack scheme nearly matches the performance of the $1/H$-portion white-box attacks, except with an additional term $307SAH^4\log(2SAT/p)/\Delta_{min}$. This additional term is dependent on how we derive the lower confidence bound. In our current paper, this bound is derived using Hoeffding-type martingale concentration inequalities. This bound can be improved by using Bernstein-type concentration inequalities. Furthermore, in our experiment results, we observe that the cost and loss of $1/H$-portion white-box attacks are about $H/\Delta_{min}$ times as much as the regret, and the performance of LCB-H black-box attacks is very close to that of the $1/H$-portion attacks, which suggests that the true performance of the designed black-box attack is very similar to the white-box attack strategy whose dependence of $H$ is only linear. Thus, we believe that our algorithm is practical when the horizon is not too long.
>
> We agree with the reviewer that the cost and loss may be large when the target policy is very close to the worst policy. This is mainly due to the fact that the influence of the action poisoning attack on the agent's observations is more indirect as compared to observation poisoning attack (please refer to our reply to reviewer AUr4' comment \#4 for more details about this point). Action poisoning can not directly change the reward $r_h$ to any desired value. If the target policy is the worst, our algorithm can not fool the agent to believe the non-target policy is worse than the target policy. In order to fool the agent, the attacker needs to change the target action to a better action or the optimal action, which may cause huge cost or loss. We believe that it will bring some ideas to build safe RL systems which are robust to the action poisoning attacks.

---

### Official Review · Reviewer_AUr4 · 2021-07-19

**Rating:** 6
**Confidence:** 4

**Summary:**

The paper presents a new attack framework on Reinforcement Learning systems that modifies the action taken by the agent without the agent noticing in order to make the agent learn the policy that the adversary wants with high probability.

**Limitations And Societal Impact:**

Yes, but would have been good to propose at least a basic defense.

**Main Review:**

Overall, the paper has good technical content but I have the issues stated below:

The threat model seems contrived and overly powerful - I am not sure how it would ever be possible for an attacker to completely swap out actions taken by an agent with completely different actions. In an MDP setting, it seems easy to identify that a less than optimal policy is in place (RL policies have the advantage of being able to learn even after deployment), and as mentioned later, so I am not convinced how broadly applicable the ideas are to the RL setting.
Also, why wouldnt such an adversary sitting between the agent and the simulator also be able to modify any observation? The authors claim that modifying observations is more powerful than actions; this is stated without any evidence (mathematical or empirical); this may not be true for some problems and some prior work restrict the modification in observations.

The white box attack is very simple and could be compressed a lot in presentation.
Perhaps more standard or complex settings for tabular episodic MDPs would help us evaluate the validity of the results more effectively than the very simple synthetic set-up.

Generally the paper could read better and has language issues, some examples include (line 21 "applicants" vs. "applications", Figure 1 contains typos ("Tiem step" vs Time step).

-------Post Response--------
I increased my score as the threat model was well explained in response.

**Time Spent Reviewing:**

3 hours

---

> ### Author Response · Authors · 2021-08-10
> **Reply to the reviewer's question**
>
> We appreciate Reviewer AUr4 for the valuable comments, and we address the reviewer's concerns as follows.
>
> 1. Comment: The threat model seems contrived and overly powerful - I am not sure how it would ever be possible for an attacker to completely swap out actions taken by an agent with completely different actions.
>
> Reply: In some applications of RL models, action decisions and reward signals may need to be transmitted over communication links. For example, in autonomous vehicles, decisions are made at central control units (CPU) and these decisions will need to be transmitted to various parts of the vehicle through communication links. When data packets containing the reward signals and action decisions etc are transmitted through the communication links, an attacker can implement adversarial attacks by intercepting and modifying these data packets. Hence, our action poisoning attack is possible.
>
> We note that the goal of this paper is not to promote action manipulation attacks. Our goal is to understand the potential risks of action manipulation attacks. Understanding the risks of different kinds of adversarial attacks on RL is essential for the safe applications of RL model and designing robust RL systems. Our study reveals that, with small costs, action manipulation attacks can indeed lead the agent to suboptimal policies. We hope that our study could lead to more follow-up work (either by ourselves or other researchers) to make RL algorithms robust to such types of attacks.
>
> 2. Comment:  In an MDP setting, it seems easy to identify that a less than optimal policy is in place (RL policies have the advantage of being able to learn even after deployment), and as mentioned later, so I am not convinced how broadly applicable the ideas are to the RL setting.
>
> Reply: Our attack is an online attack, i.e, the attacker attacks during the victim's online learning process. In RL problems, the reward functions and the state transferring are generally not deterministic. Hence, if the agent does not know the true underlying MDP, which is the case in most RL applications, it is not straightforward to detect such attacks. Even in deployment, using the attack strategy discussed in the paper, the target policy chosen by the attacker is optimal from the agent's perspective.
>
> It is indeed important to investigate how to defend against such types of attacks. We hope that the paper can inspire follow-up work that can detect and mitigate such attacks.
>
> 3. Comment: Why wouldn't such an adversary sitting between the agent and the simulator also be able to modify any observation?
>
> Reply: Yes, the adversary may also be able to modify the observation. Indeed, there are already existing works that study the impact of observation modification attacks. We believe that studying every different kind of attack is essential for the safe applications of RL model. Thus, in this paper, we focus on action poisoning attacks in which the adversary only modifies the action signal. In our paper, we showed that by only attacking actions, the adversary can efficiently force the agent to choose actions according to a target policy. The results expose a security threat of the action poisoning attacks on RL and may motivate us to design a robust RL system that can detect and mitigate such attacks.
>
> 4. Comment: The authors claim that modifying observations is more powerful than actions; this is stated without any evidence (mathematical or empirical); this may not be true for some problems and some prior work restrict the modification in observations.
>
> Reply: The agent makes decision based on the observations, i.e., the trajectory at each episode $\\{ s_1, a_1, r_1, \dots, s_H, a_H, r_H, s_{H+1} \\}$. Modifying observations can change the reward $r_h$ to any value or change the observed state $s_h$ to a specific state, while the action poisoning only can indirectly impact the observations. If the attacker can change the observations as in the reward poisoning setting, it is easy to design an attack strategy without needing to learn the value of the target policy and other quantities related to the underlying MDP. In particular, the attacker can simply change the reward of any non-target action to 0 when the reward function is bounded by $[0,1]$ or any value lower than the lower bound of all rewards. Such an attacker can force any no-regret RL to pull action according to the target policy by spending sublinear costs. Certainly, if there is an attack power constraint, e.g. the manipulated reward was constrained within a $\epsilon$-ball centered at the unpoisoned reward, the observation poisoning is more restricted and less powerful. We will make it more clear in the modified version.
>
> 5. Comment: The white box attack is very simple and could be compressed a lot in presentation. Perhaps more standard or complex settings for tabular episodic MDPs would help us evaluate the validity of the results more effectively than the very simple synthetic set-up. Generally the paper could read better and has language issues, some examples include (line 21 "applicants" vs. "applications", Figure 1 contains typos ("Tiem step" vs Time step).
>
> Reply: We will compress the part of the white-box attack and fix the typo in the modified version.
>
> Regarding the simulation results, it is indeed true that our experimental setting is relatively simple. However, we note that our paper is mainly a theoretical paper with theoretical performance guarantees. We will add some simulation results of more standard or complex settings in the revised version.

---

### Decision · Program_Chairs · 2021-09-27

**Decision:**

Accept (Poster)

**Comment:**

This paper proposes a new, action poisoning based attack scheme against tabular episodic RL algorithms. The authors investigate this new attack scheme in both white-box and black-box settings. For the white-box setting, the authors propose a simple attack mechanism. For the black-box case, the authors  develop a new attack that provably approximately as good as the white-box attack.

I agree with most reviewers that the theoretical findings of this paper are very useful to understand the limitations of RL algorithms. It is worth mentioning thought that the reviewers still have some concerns regarding the scalability of the proposed attacks to non-tabular settings, which is more realistic than the tabular one studied in this paper. Despite this concern, I can conclude that the reviewers did not find any reasons to reject this paper (and neither did I). In fact, all of us are in favour of acceptance. Hence I recommend this paper to be accepted as a poster.